# WEAKLY-SUPERVISED NEURO-SYMBOLIC IMAGE MANIPULATION VIA MULTI-HOP COMPLEX INSTRUCTIONS

## ABSTRACT

We are interested in *image manipulation via natural language text* – a task that is extremely useful for multiple AI applications but requires complex reasoning over multi-modal spaces. Recent work on *neuro-symbolic approaches* e.g., The Neuro Symbolic Concept Learner (NSCL) (Mao et al., 2019) has been quite effective for solving Visual Question Answering (VQA) as they offer better *modularity*, *interpretability*, and *generalizability*. We extend NSCL for the image manipulation task and propose a solution referred to as NEUROSIM. Previous work either requires *supervised training data* in the form of manipulated images or can only deal with very simple reasoning instructions over single object scenes. In contrast, NEUROSIM can perform *complex multi-hop reasoning* over *multi-object scenes* and only requires *weak supervision* in the form of annotated data for VQA. NEUROSIM parses an instruction into a symbolic program, based on a Domain Specific Language (DSL) comprising of object attributes and manipulation operations, that guides the manipulation. We design neural modules for manipulation, as well as *novel loss functions* that are capable of testing the correctness of manipulated object and scene graph representations via query networks trained merely on VQA data. An image decoder is trained to render the final image from the manipulated scene graph. Extensive experiments demonstrate that NEUROSIM, without using target images as supervision, is highly competitive with SOTA baselines that make use of supervised data for manipulation.

## 1 INTRODUCTION

The last decade has seen a significant growth in application of *neural models* to a variety of tasks including those in computer vision (Chen et al., 2017; Krizhevsky et al., 2012), NLP (Wu et al., 2016), robotics and speech (Yu & Deng, 2016). It has been observed that these models often lack interpretability (Fan et al., 2021), and may not always be well suited to handle complex reasoning tasks (Dai et al., 2019). On the other hand, *classical AI systems* can seamlessly perform complex reasoning in an interpretable manner due to their *symbolic representation* (Pham et al., 2007; Cai & Su, 2012). But these models are often found lacking in their ability to handle low level representations and be robust to noise. A natural question then arises: *Can we design models which capture the best of both these paradigms?* The answer lies in the recent development of *Neuro-Symbolic models* (Dong et al., 2019; Mao et al., 2019; Han et al., 2019) which combine the power of (purely) neural with (purely) symbolic representations. An interesting sub-class of these models work with a finite sized domain specification language (DSL) and make use of deep networks to learn neural representations of the concepts specified in the DSL. The learned representations are then used for performing downstream reasoning via learning of symbolic programs. This line of work was first popularized by Andreas et al. (2016); Hu et al. (2017); Johnson et al. (2017a), followed by Mao et al. (2019), who look at the task of Visual Question Answering (VQA), and other follow-up works such as learning meta-concepts (Han et al., 2019). Studies (Andreas et al., 2016; Hu et al., 2017; Mao et al., 2019) have shown that these models have several desirable properties such as *modularity, interpretability,* and *improved generalizability*.

Motivated by the above, our aim is to build neuro-symbolic models for the task of *weakly supervised manipulation of images comprising multiple objects, via complex multi-hop natural language instructions.* Existing work includes weakly supervised approaches (Nam et al., 2018; Li et al., 2020) that require textual descriptions of images during training and are limited to very simple scenes (or

instructions). Supervised approaches (Zhang et al., 2021; El-Nouby et al., 2019), though capable of handling multiple objects and complex multi-hop instructions, require explicit annotations in the form of target manipulated images; ref. Section 2 for a survey. We are interested in a weakly supervised solution that only makes use of data annotated for VQA, avoiding the high cost of getting supervised annotations, in the form of target manipulated images. Our key intuition is: this task can be solved by simply querying the manipulated representation without ever explicitly looking at the target image.

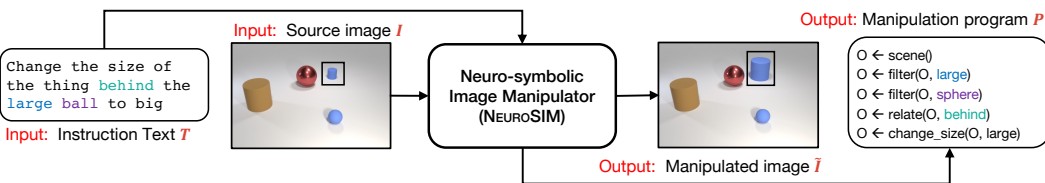

Figure 1: The problem setup.

Our solution builds on Neuro-Symbolic Concept Learner (NSCL) proposed by Mao et al. (2019) for solving VQA. We extend this work to incorporate the notion of manipulation operations such as *change*, *add,* and *remove* objects in a given image. As one of our main contributions, we design novel neural modules and a training strategy that just uses VQA annotations as weakly supervised data for the task of image manipulation. The neural modules are trained with the help of *novel loss functions* that measure the faithfulness of the manipulated scene and object representations by accessing a separate set of *query networks*, interchangeably referred to as *quantization networks*, trained just using VQA data. The manipulation takes place through interpretable programs created using primitive neural and symbolic operations from a Domain Specific Language (DSL). Separately, a network is trained to render the image from a scene graph representation using a combination of $L_1$ and adversarial losses as done by Johnson et al. (2018). The entire pipeline is trained without any intermediate supervision. We refer to our system as Neuro-Symbolic Image Manipulator (NEUROSIM). Figure 1 shows an example of I/O pair for our approach.

For our experiment purposes, we extend CLEVR (Johnson et al., 2017b), a benchmark dataset for VQA, to incorporate manipulation instructions and create a dataset referred to as *Complex Image Manipulation via Natural Language Instructions* (CIM-NLI). We will release this dataset publicly post acceptance. Our evaluation on CIM-NLI dataset shows that, despite being weakly supervised. we are highly competitive or improve upon state-of-the-art supervised approaches (Zhang et al., 2021; El-Nouby et al., 2019) for this task, generalize well to scenes with more objects, and specifically perform well on instructions which involve multi-hop reasoning.

## 2 RELATED WORK

Table 1 categorizes the related work across three broad dimensions - *problem setting*, *task complexity*, and *approach*. The problem setting comprises of two sub-dimensions: i) supervision type - *self, direct,* or *weak*, ii) instruction format - *text or UI-based*. The task complexity comprises of following sub-dimensions: ii) scene complexity – *single* or *multiple objects*, ii) instruction complexity - *zero or multi-hop instructions*, iii) kinds of manipulations allowed - *add, remove,* or *change*. Finally, approach consists of the following sub-dimensions: i) model – *neural* or *neuro-symbolic* and ii) whether symbolic program is generated on the way or not.

Dong et al. (2017), TAGAN (Nam et al., 2018), and ManiGAN (Li et al., 2020) are close to us in terms of the problem setting. These manipulate the source image using a GAN-based encoder-decoder architecture. Their weak supervision differs from ours – We need VQA annotation, they need captions or textual descriptions. The complexity of their natural language instructions is restricted to 0-hop. Most of their experimentation is limited to single (salient) object scenes, and it is unclear how these strategies would perform with multi-object situations with intricate relationships. Lastly, while our approach requires only an explicit manipulation (delta) command during inference, existing approaches require partial target image description, and it is unclear how their method can be extended to the task where only the delta is given.

In terms of task complexity, the closest to us are approaches such as TIM-GAN (Zhang et al., 2021), GeNeVA (El-Nouby et al., 2019), which build an encoder decoder architecture and work with a latent representation of the image as well as the manipulation instruction. They require explicit annotations in terms of manipulated images during training. We argue that this can require a significant more

| Prior Work | Problem Setting | | Task Complexity | | | Approach | |
|---|---|---|---|---|---|---|---|
| | ST | IF | SC | IC | Operations | Model | Program |
| SIMSG | SS | UI | MO† | N/A | change, remove, add | N | ✗ |
| PGIM | DS | N/A | MO†* | N/A | change (image level) | NS | ✓ |
| GeNeVA | DS | Text# | MO | MH | add | N | ✗ |
| TIM-GAN | DS | Text# | MO† | ZH | change, remove, add | N | ✗ |
| Dong et. al | WS | Text# | SO† | ZH | change | N | ✗ |
| TAGAN | WS | Text# | SO† | ZH | change | N | ✗ |
| ManiGAN | WS | Text# | SO† | ZH | change | N | ✗ |
| NEUROSIM (ours) | WS | Text | MO | MH | change, remove, add | NS | ✓ |

Table 1: Comparison of Prior Work. Abbreviations (column titles) ST:= Supervision Type, IF:= Instruction Format, SC:= Scene Complexity, IC:=Instruction Complexity. Abbreviations (column values) SS:=Self Supervision, DS:=Direct Supervision, WS:=Weak Supervision, #: Human Written, MO:= Multiple Objects, MO*:= Multiple Objects with Regular Patterns, SO:= Single Object, †: Natural Images, N/A:= Not applicable, MH:=Multi-Hop, ZH:=Zero-Hop, N:= Neural, NS:= Neuro-Symbolic, ✓:= Yes, ✗:= No. See Section 2 for more details.

annotation effort, compared to our weak supervision setting, where we only need visual question answer annotations. Unlike us, these approaches work with purely neural models, and as shown in our experiments, their performance is heavily dependent on the amount of data available for training.

In terms of technique, the closest to our work are neuro-symbolic approaches for VQA such as NSVQA (Yi et al., 2018), NSCL (Mao et al., 2019), Neural Module Networks (Andreas et al., 2016) and its extensions (Hu et al., 2017; Johnson et al., 2017a). Clearly, while the modeling approach is similar and consists of constructing latent programs, the desired task are different in two cases. Our work extends the NSCL approach for the task of automated image manipulation.

A related task is *text guided image retrieval*, where goal is to retrieve (not manipulate) an image from the database complying with the changes asked for in the input instruction (Vo et al., 2019; Chen et al., 2020). Another line of research (Jiang et al., 2021; Shi et al., 2021) deals with editing global features, such as brightness, contrast, etc., instead of object level manipulations like in our case. Recent works (Ramesh et al., 2022; Saharia et al., 2022) on text to image generation using diffusion models trained on massive (image, caption) data, are capable of generating photorealistic images given text. These also have the capability of editing images e.g. using text-diffs (Ramesh et al., 2022) but require captions for input images. Further, it is unclear how to extend this line of work to language guided complex image manipulation settings where multi-hop reasoning may be required; preliminary studies (Marcus et al., 2022) have highlighted their shortcomings in terms of compositional reasoning and dealing with relations.

# 3 NEUROSIM: NEURO-SYMBOLIC IMAGE MANIPULATOR

## 3.1 MOTIVATION AND ARCHITECTURE OVERVIEW

The key motivation behind our approach comes from the following hypothesis: consider a learner $L$ (e.g., a neural network or the student in Fig 2) with sufficient capacity trying to achieve the task of manipulation over Images $I$. Further, let

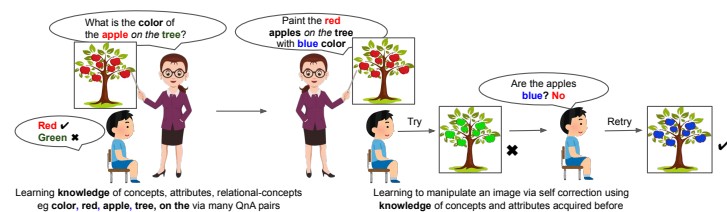

Figure 2: Motivating example for our approach.

each image be represented in terms of its properties, or properties of its constituent parts (e.g. objects like apple, leaf, tree as shown in Fig 2), where each property comes from a finite set $S$ e.g, attributes of objects in an image. Let the learner be provided with the prior knowledge (for e.g. through Question Answering as in Fig 2) about properties (e.g., color) and their possible values (e.g., red). Then, in order to learn the task of manipulation, it suffices to provide the learner with a *query network*, which given a manipulated image $\tilde{I}$ constructed by the learner via command $C$, can correctly answer questions (i.e. query) about the desired state of various properties of the constituents of the image

$\tilde{I}$. The query network can be internal to the learner (e.g., the student in Fig 2 can query himself for checking the color of apples in the manipulated image). The learner can query repeatedly until it learns to perform the manipulation task correctly. Note that the learner does not have access to the supervised data corresponding to triplets of the form $(I_s, C, I_f)$, where $I_s$ is the starting image, $C$ is the manipulation command, and $I_f$ is the resulting final image, for the task of manipulation. Inspired by this, we set out to test this hypothesis by building a model capable of manipulating images, without target images as supervision.

Figure 3 captures a high level architecture of the proposed NEUROSIM pipeline. NEUROSIM allows manipulating images containing multiple objects, via complex natural language instructions. Similar to Mao et al. (2019), NEUROSIM assumes the availability of a *domain-specific language* (DSL) for parsing the instruction text $T$ into an executable program $P$. NEUROSIM is capable of handling *addition, removal,* and *change* operations over image objects. It reasons over the image for locating where the manipulation needs to take place followed by carrying out the manipulation operation. The first three modules, namely *i) visual representation network, ii) semantic parser,* and *iii) concept quantization network* are suitably customized from the NSCL and trained as required for our purpose. In what follows, we describe the design and training mechanism of NEUROSIM.

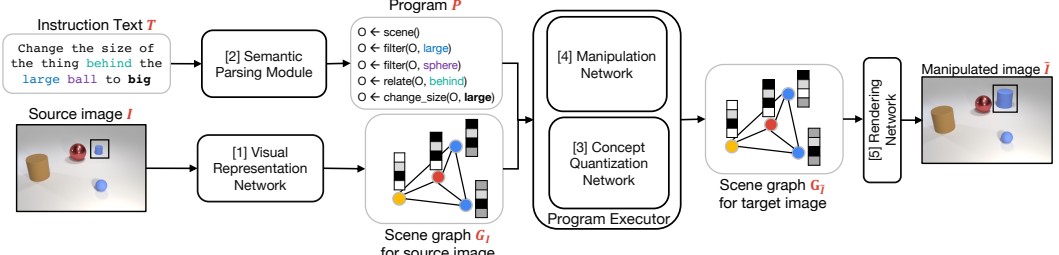

Figure 3: High level architecture of NEUROSIM.

## 3.2 MODULES INHERITED FROM NSCL

**1] Visual Representation Network:** Given input image $I$, this network converts it into a scene graph $G_I = (N, E)$. The nodes $N$ of this scene graph are object embeddings and the edges $E$ are embeddings capturing relationship between pair of objects (nodes). Node embeddings are obtained by passing the bounding box of each object (along with the full image) through a ResNet-34 (He et al., 2016b). Edge embeddings are obtained by concatenating the corresponding object embeddings.

**2] Semantic Parsing Module:** The input to this module is a manipulation instruction text $T$ in natural language. Output is a *symbolic program $P$* generated by parsing the input text. The symbolic programs are made of operators, that are part of our DSL (Specified in Appendix Section A).

**3] Concept Quantization Network:** Any object in an image is defined by the set of *visual attributes* ($A$), and set of symbolic values ($S_a$) for each attribute $a \in A$. E.g., attributes can be *shape, size,* etc. Different symbolic values allowed for an attribute are also known as *concepts*. E.g., $S_{\text{color}} = \{red, blue, green, \dots\}$. Each visual attribute $a \in A$ is implemented via a separate neural network $f_a(\cdot)$ which takes the object embedding as input and outputs the attribute value for the object in a *continuous (not symbolic)* space. Let $f_{\text{color}} : \mathbb{R}^{d_{\text{obj}}} \to \mathbb{R}^{d_{\text{attr}}}$ represent a neural network for the *color* attribute and consider $o \in \mathbb{R}^{d_{\text{obj}}}$ as the object embedding. Then, $v_{\text{color}} = f_{\text{color}}(o) \in \mathbb{R}^{d_{\text{attr}}}$ is the embedding for the object $o$ pertaining to the color attribute. Each symbolic concept $s \in S_a$ for a particular attribute $a$ (e.g., different colors) is also assigned a respective embedding in the same continuous space $\mathbb{R}^{d_{\text{attr}}}$. Such an embedding is denoted by $c_s$. These concept embeddings are initialized at random, and later on fine tuned during training. An attribute embedding (e.g. $v_{\text{color}}$) can be compared with the embeddings of all the concepts (e.g., $c_{\text{red}}, c_{\text{blue}}$, etc.) using cosine similarity, for the purpose of concept quantization of objects.

**Training for VQA:** As a first step, we train all the above three modules via a curriculum learning process (Mao et al., 2019). Semantic parser is trained jointly with the concept quantization networks for generating programs corresponding to the question texts coming from the VQA dataset. The corresponding output programs are composed of primitive operations coming from the DSL (e.g. *filter, count,* etc.) and does not include constructs related to manipulation operations. This trains the first three modules with high accuracy on the VQA task.

### 3.3 Novel Modules and Training Procedure for NeuroSIM

NeuroSIM training starts with three sub-modules trained on the VQA task as described in Section 3.2. Next, we extend the original DSL to include three additional functional sub-modules within semantic parsing module, namely *add*, *remove*, and *change*. Refer to appendix section A for details on the DSL. We now reset the semantic parsing module and train it again from scratch for generating programs corresponding to *image manipulation instruction text $T$*. Such a program is subsequently used by the downstream pipeline to reason over the scene graph $G_I$ and manipulate the image. In this step, the semantic parser is trained using an off-policy program search based REINFORCE (Williams, 1992) algorithm. Unlike the training of semantic parser for the VQA task, in this step, we *do not* have any final *answer like* reward supervision for training. Hence, we resort to a weaker form of supervision. In particular, consider an input instruction text $T$ and set of all possible manipulation program templates $\mathbb{P}_t$ from which one can create any actual program $P$ that is executable over the scene graph of the input image. For a program $P \in \mathbb{P}_t$, our reward is positive if this program $P$ selects any object (or part of the scene graph) to be sent to the manipulation networks (change/add/remove). For e.g., consider the program *change(filter(scene()))*, if after executing *filter(scene())*, we do not get even a single object selected, then we give a negative reward, signifying that this program cannot be correct, else we give a positive reward. Incorrect programs can also lead to object (objects) being selected for manipulation, which is why this is a weak supervision. See Appendix C for more details.

Once semantic parser is retrained, we clamp the first three modules and continue using them for the purpose of parsing instructions and converting images into their scene graph representations. Scene graphs are manipulated using our novel module called *manipulation network* which is describe next.

**4] Manipulation Network:** This is our key module responsible for carrying out the manipulation operations. We allow three kinds of manipulation operations – *add, remove,* and *change*. Each of these operations are a composition of a quasi-symbolic and symbolic operation. A symbolic operation corresponds to a function that performs the required structural changes (i.e. addition/deletion of a node or an edge) in the scene graph $G_I$ against a given instruction. A quasi-symbolic operations is a dedicated neural network that takes the relevant part of $G_I$ as input and computes new representations of nodes and edges that are compatible with the changes described in the parsed instruction.

**(a) Change Network:** For each visual attribute $a \in A$ (e.g. *shape, size, ...*), we have a separate *change neural network* that takes the pair of *(object embedding, embedding of the changed concept)* as input and outputs the embedding of the *changed* object. This is the quasi-symbolic part for the change function, while the symbolic part is an identity mapping. For e.g., let $g_{\text{color}} : \mathbb{R}^{d_{\text{obj}} + d_{\text{attr}}} \rightarrow \mathbb{R}^{d_{\text{obj}}}$ represent the neural network that changes the *color* of an object. Consider $o \in \mathbb{R}^{d_{\text{obj}}}$ as the object embedding and $c_{\text{red}} \in \mathbb{R}^{d_{\text{attr}}}$ as the concept embedding for the *red* color, then $\widetilde{o} = g_{\text{color}}(o; c_{\text{red}}) \in \mathbb{R}^{d_{\text{obj}}}$ represents the changed object embedding, whose color would be *red*. After applying the change neural network, we obtain the changed representation of the object $\widetilde{o} = g_a(o; c_{s_a^*})$, where $s_a^*$ is the desired changed value for the attribute $a$. This network is trained using following losses.

$$\ell_a = -\sum_{\forall s \in S_a} \mathbb{I}_{s = s_a^*} \ \log\left[p(h_a(\widetilde{o}) = s)\right] \tag{1}$$

$$\ell_{\overline{a}} = -\sum_{\forall a' \in A, a' \neq a} \sum_{\forall s \in S_{a'}} p(h_{a'}(o) = s) \log\left[p(h_{a'}(\widetilde{o}) = s)\right] \tag{2}$$

where, $h_a(x)$ gives the concept value of the attribute $a$ (in symbolic form $s \in S_a$) for the object $x$. The quantity $p(h_a(x) = s)$ denotes the probability that the concept value of the attribute $a$ for the object $x$ is equal to $s$ and is given as follows $p(h_a(x) = s) = \exp^{dist(f_a(x), c_s)} / \sum_{\widetilde{s} \in S_a} \exp^{dist(f_a(x), c_{\widetilde{s}})}$ where, $dist(a, b) = (a^\top b - t_2)/t_1$ is the shifted and scaled cosine similarity, $t_1, t_2$ being constants. The first loss term $\ell_a$ penalizes the model if the (symbolic) value of the attribute $a$ for the manipulated object is different from the desired value $s_a^*$ in terms of probabilities. The second term $\ell_{\overline{a}}$, on the other hand, penalizes the model if the values of any of the other attributes $a'$, deviate from their original values. Apart from these losses, we also include following additional losses.

$$\ell_{\text{cycle}} = \|o - g_a(\widetilde{o}; c_{\text{old}})\|_2; \ \ \ell_{\text{consistency}} = \|o - g_a(o; c_{\text{old}})\|_2 \tag{3}$$

$$\ell_{\text{objGAN}} = -\sum_{o' \in O}\left[\log D(o') + \log(1 - D(g_a(o'; c)))\right] \tag{4}$$

where $c_{old}$ is the original value of the attribute $a$ of object $o$, before undergoing change. Intuitively the first loss term $\ell_{\text{cycle}}$ says that, changing an object and then changing it back should result in the same object. The second loss term $\ell_{\text{consistency}}$ intuitively means that changing an object $o$ that has value $c_{old}$

for attribute $a$, into a new object with the same value $c_{old}$, should not result in any change. These additional losses prevent the change network from changing attributes which are not explicitly taken care of in earlier losses (1) and (2). For e.g., rotation or location attributes of the objects that are not part of our DSL. We also impose an adversarial loss $\ell_{objGAN}$ to ensure that the new object embedding $\widetilde{o}$ is from the same distribution as real object embeddings. See Appendix C for more details.

**(b) Remove Network:** The remove network takes the scene graph $G_I$ of the input image and removes the subgraph from $G_I$ that contains the nodes (and incident edges) corresponding to the object(s) that need to be removed, and returns a new scene graph $G_{\widetilde{I}}$ which is reduced in size. The quasi-symbolic function for the remove network is identity.

**(c) Add Network:** For adding a new object into the scene, *add network* requires the symbolic values of different attributes, say $\{s_{a_1}, s_{a_2}, \ldots, s_{a_k}\}$, for the new object, e.g., {red, cylinder, ...}. It also requires the spatial relation $r$ (e.g. RightOf) of the new object with respect to an existing object in the scene. The add function works by first predicting the object (node) embedding $\widetilde{o}_{new}$ for the object to be added, followed by predicting edge embeddings for new edges incident on the new node. New object embedding is obtained as follows: $\widetilde{o}_{new} = g_{addObj}(\{c_{s_{a_1}}, c_{s_{a_2}}, \cdots, c_{s_{a_k}}\}, o_{rel}, c_r)$ where, $o_{rel}$ is the object embedding of an existing object, relative to which new object's position $r$ is specified. After this, for each existing objects $o_i$ in the scene, an edge $\widetilde{e}_{new,i}$ is predicted between the newly added object $\widetilde{o}_{new}$ and existing object $o_i$ in following manner: $\widetilde{e}_{new,i} = g_{addEdge}(\widetilde{o}_{new}, o_i)$. Functions $g_{addObj}(\cdot)$ and $g_{addEdge}(\cdot)$ are quasi-symbolic operations. Symbolic operations in *add network* comprises adding the above node and the incident edges into the scene graph.

The *add network* is trained in a *self-supervised* manner. For this, we pick a training image and create it's scene graph. Next, we randomly select an object $o$ from this image and quantize it's concepts, along with a relation with any other object $o_i$ in the same image. We then use our *remove* network to remove this object $o$ from the scene. Finally, we use the quantized concepts and the relation that were gathered above and add this object $o$ back into the scene graph using $g_{addObj}(\cdot)$ and $g_{addEdge}(\cdot)$. Let the embedding of the object after adding it back is $\widetilde{o}_{new}$. Training losses for this network are as follows:

$$\ell_{concepts} = -\sum_{j=1}^{k} \log\big(p(h_{a_j}(\widetilde{o}_{new}) = s_{a_j})\big); \; \ell_{relation} = -\log(p(h_r(\widetilde{o}_{new}, o_i) = r)) \quad (5)$$

$$\ell_{objSup} = \|o - \widetilde{o}_{new}\|_2; \; \ell_{edgeSup} = \sum_{i \in O} \|e_{old,i} - \widetilde{e}_{new,i}\|_2 \quad (6)$$

$$\ell_{edgeGAN} = -\sum_{\forall i \in O} [\log D(\{o; e_{old,i}; o_i\}) + \log(1 - D(\{\widetilde{o}_{new}; \widetilde{e}_{new,i}; o_i\}))] \quad (7)$$

where $s_{a_j}$ is the required (symbolic) value of the attribute $a_j$ for the original object $o$, and $r$ is the required relational concept. $O$ is the set of the objects in the image, $e_{old,i}$ is the edge embedding for the edge between original object $o$ and its neighboring object $o_i$. Similarly, $\widetilde{e}_{new,i}$ is the corresponding embedding of the same edge but after when we have (removed + added back) the original object. The loss terms $\ell_{concepts}$ and $\ell_{relation}$ ensure that the added object comprises desired values of attributes and relation, respectively. Since we had first removed and then added the object back, we already have the original edge and object representation, and hence we use them in loss terms given in equation 6. We use adversarial loss equation 7 for generating real (object, edge, object) triples and also a loss similar to equation 4 for generating real objects. For optimizing the generator in eq. equation 4 equation 7 modified GAN loss (Goodfellow et al., 2014) is used.

### 3.4 IMAGE RENDERING FROM SCENE GRAPH

**5] Rendering Network:** Design and training methodology for this module closely follows Johnson et al. (2018). We take multiple images $\{I_1, I_2 \cdots I_n\}$ and generate their scene graph $\{G_{I_1}, G_{I_2} \cdots G_{I_n}\}$ using the *visual representation network* described earlier. Each of these scene graphs is then processed by a graph convolutional network and passed through a mask regression network followed by a box regression network to generate a coarse 2-dimensional structure (scene layout). A Cascaded Refinement Network (CRN) (Chen & Koltun, 2017) is then employed to generate an image from the the scene layout. A *min-max adversarial training procedure* is used to generate realistic images as formulated in GAN (Goodfellow et al., 2014), using two discriminators – i) A patch-based image discriminator that ensures the quality of overall image, and ii) An object discriminator that ensures the quality of object appearance.

## 4 EXPERIMENTS

**Datasets:** Among the existing datasets, CSS (Vo et al., 2019) contains simple 0-hop instructions and is primarily designed for the text guided image retrieval task. CRIR (Chen et al., 2020) extends CSS to include multi-hop instructions but is not open source. Other datasets such as i-CLEVR (El-Nouby et al., 2019) and CoDraw are designed for iterative image editing (El-Nouby et al., 2019). i-CLEVR contains only "add" instructions and CoDraw doesn't contain multi-hop instructions. Hence we created our own *multi-object multi-hop instruction* based image manipulation dataset, referred to as CIM-NLI. This dataset was generated with the help of CLEVR toolkit (Johnson et al., 2017b) – details of the generation process are described in the Appendix B. CIM-NLI consists of (Source image $I$, Instruction text $T$, Target image $\widetilde{I}^*$) triplets. The dataset contains a total of $18K, 5K, 5K$ unique images and $54K, 14K, 14K$ instructions in the *train, validation and test* splits respectively. Refer Appendix B for more details about the dataset splits.

**Baselines:** Weakly supervised baselines (Li et al., 2020; Nam et al., 2018) for this task are designed for a problem setting different from ours – single salient object scenes, simple 0-hop instructions (Refer Section 2 for details). Further, they require paired images and their textual descriptions as annotations. We, therefore, do not compare with them in our experiments. Instead, we compare our model with purely supervised approaches such as TIM-GAN (Zhang et al., 2021) and GeNeVA (El-Nouby et al., 2019). In order to make a fair and meaningful comparison between the two kinds (supervised and weakly-supervised) approaches, we carve out the following set-up. Assume the cost required to create one single annotated example for image manipulation task be $\alpha_m$ while the corresponding cost for the VQA task be $\alpha_v$. Let $\alpha = \alpha_m/\alpha_v$. Let $\beta_m$ be the number of annotated examples required by a supervised baseline for reaching a performance level of $\eta_m$ on the image manipulation task. Similarly, let $\beta_v$ be the number of annotated VQA examples required to train NEUROSIM to reach the performance level of $\eta_v$. Let $\beta = \beta_m/\beta_v$. We are interested in figuring out the range of $\beta$ for which performance of our system ($\eta_v$) is at least as good as the baseline ($\eta_m$). Correspondingly we can compute the ratio of the labelling effort required, i.e., $\alpha * \beta$, to reach these performance levels. If $\alpha * \beta > 1$, our system achieves the same or better performance, with lower annotation cost. See Appendix F, G for computational resources and hyperparameters respectively.

**Evaluation Metrics:** For evaluation on image manipulation task, we use two metrics - i) *FID*, ii) Recall@$k$. FID (Heusel et al., 2017) measures the realism of the generated images. Recall@$k$ measures the semantic similarity of gold manipulated image $\widetilde{I}^*$ and system produced manipulated image $\widetilde{I}$. For computing Recall@$k$, we use $\widetilde{I}$ as a query and retrieve images from a corpus comprising the entire test set (gold manipulated images) and the source image $I$ corresponding to $\widetilde{I}$. similar to Zhang et al. (2021), the query image and the corpus images are embedded into a latent space through an *autoencoder* trained on CLEVR dataset. Cosine similarity is used for ranking retrieved images.

| Method | $\beta = 0.054$ | | | $\beta = 0.07$ | | | $\beta = 0.1$ | | | $\beta = 0.2$ | | | $\beta = 0.54$ | | |
| | FID | $R1$ | $R3$ | FID | $R1$ | $R3$ | FID | $R1$ | $R3$ | FID | $R1$ | $R3$ | FID | $R1$ | $R3$ |
|---|---|---|---|---|---|---|---|---|---|---|---|---|---|---|---|
| GeNeVA | 22.0 | 6.6 | 58.7 | – | – | – | – | – | – | – | – | – | 10.3 | 4.6 | 64.4 |
| TIM-GAN | 4.8 | 31.9 | 74.2 | 4.4 | 32.6 | 80.0 | 4.3 | 38.5 | 82.5 | 4.9 | 47.4 | 86.4 | 4.0 | 58.1 | 90.2 |
| NEUROSIM | 13.8 | 45.3 | 65.5 | 13.7 | 45.8 | 66.7 | 14.0 | 45.6 | 66.7 | 14.1 | 45.6 | 67.9 | 13.8 | 45.5 | 66.7 |

Table 2: Performance comparison of NEUROSIM with TIM-GAN and GeNeVA with varying $\beta$ levels. The '-' entries for GeNeVA were not computed due to excessive training time; it's performance is low even when using full data. We always use $100K$ VQA examples (5K Images, 20 questions per image) for our weakly supervised training. $R1, R3$ correspond to Recall@1,3 respectively. FID: lower is better; Recall: higher is better.

### 4.1 PERFORMANCE WITH VARYING DATASET SIZE

Table 2 compares the performance of NEUROSIM system with TIM-GAN and GeNeVA with varying levels of $\beta$ on CIM-NLI. Despite being weakly supervised, NEUROSIM performs significantly better than both the baselines for $\beta \leq 0.1$ ( alternatively $\alpha \geq 10$) and very close to its closest competitor for $\beta = 0.2$ ( alternatively $\alpha = 5$), using the $R@1$ performance metric. This clearly demonstrates the strength of our approach in learning to manipulate while only making use of VQA annotations. We hypothesize that, in most cases, NEUROSIM will be preferable since, we expect the cost of

annotating an output image for manipulation to be significantly higher than the cost of annotating a VQA example. FID scores for NEUROSIM could potentially be improved by a doing a joint training of VQA module along with image decoder loss, and is a direction for future work.

## 4.2 PERFORMANCE WITH INCREASING REASONING HOPS

Table 3 compares baselines with NEUROSIM for performance over instructions requiring 0-hop versus multi-hop ($1-3$ hops) reasoning. When dealing with multi-hop instructions, we see a massive drop of $14.8$ and $7.8$ points in the performance of TIM-GAN trained on $10\%$ ($5.4K$ data points) and Full ($54K$ data points) CIM-NLI data respectively. NEUROSIM trained on $10\%$ data, *without output image supervision*, sees a performance drop of only $1.5$ points implying that it is much better at handling the complex reasoning involved.

| Method | Hops | | |
|---|---|---|---|
| | $ZH$ | $MH$ | |
| GeNeVA ($54K$) | 6.3 | 6.4 | (+0.1) |
| GeNeVA ($5.4K$)) | 8.5 | 9.9 | (+1.4) |
| TIM-GAN ($54K$) | 84.0 | 76.2 | (-7.8) |
| TIM-GAN ($5.4K$) | 56.4 | 41.6 | (-14.8) |
| NEUROSIM ($5.4K$) | 64.5 | 63.0 | (-1.5) |

Table 3: $R1$ results for 0-hop (ZH) vs multi-hop (MH) instruction guided image manipulation.

## 4.3 ZERO-SHOT GENERALIZATION TO LARGER SCENES

We developed another dataset called CIM-NLI-LARGE, consisting of scenes having $10-13$ objects (See Appendix B for details). We study the combinatorial generalization ability of NEUROSIM and the baselines when the models are trained on CIM-NLI containing scenes with $3-8$ objects only and evaluated on CIM-NLI-LARGE. Table 5 captures such a comparison. NEUROSIM does significantly better than TIM-GAN and GeNeVA trained on $10\%$ ($5.4K$ data points) of CIM-NLI data for e.g. it improves over TIM-GAN $R1$ score by 33.5 points. NEUROSIM nearly matches TIM-GAN's performance trained on full CIM-NLI data. This demonstrates the superior generalization capabilities of our weakly supervised model compared to supervised baselines.

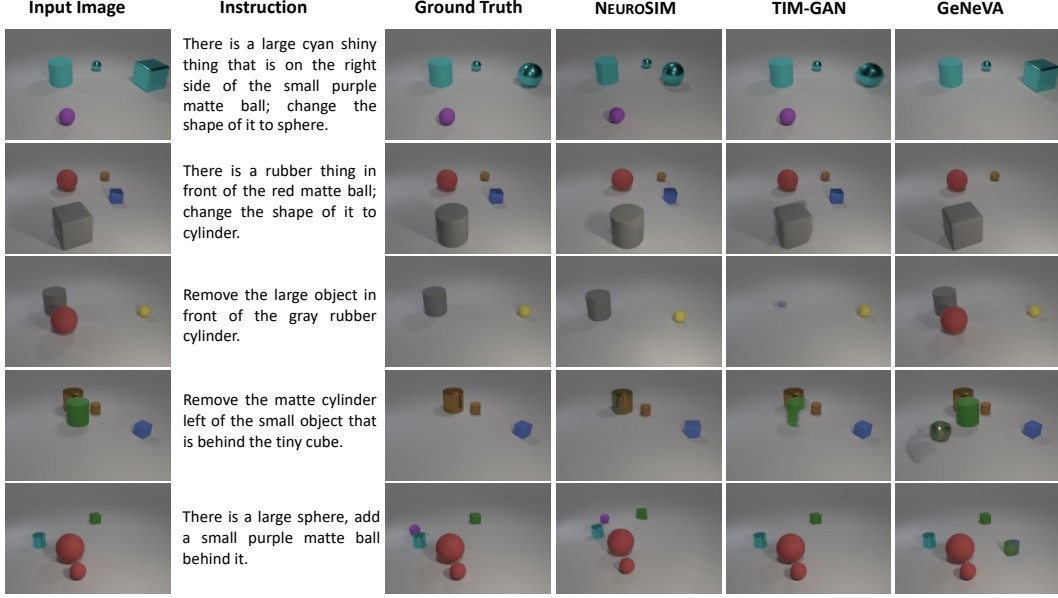

Figure 4: Visual comparison of NEUROSIM results with TIM-GAN and GeNeVA.

## 4.4 QUALITATIVE ANALYSIS AND INTERPRETABILITY

Figure 4 shows anecdotal examples for visually comparing NEUROSIM with baselines. Note, GeNeVA either performs the wrong operation on the image (row #1, 2, 4, and 5) or simply copies the input image to output without any modifications (row #3). TIM-GAN often makes semantic errors which show its lack of reasoning – for example, removing the wrong objects in row #3. Compared to baselines, NEUROSIM produces semantically more meaningful image manipulation. NEUROSIM can easily recover occluded objects (row #4). All models make rendering errors such as *partial removal of objects, shape distortion* (rows #2, 4, and 5). More results are in Section H of appendix.

NEUROSIM produces interpretable output programs, showing the steps taken by the model to edit the images. Some examples and errors are shown in Appendix J. This highlights the ease of detecting failures of NEUROSIM, which is not possible with neural baselines.

### 4.5 HUMAN EVALUATION

For the human evaluation study, in each instance, we provided evaluators with four images (1) input image, (2) ground-truth image, (3) manipulated image generated by NEUROSIM (5.4K), and (4) manipulated image generated by TIM-GAN (54K). Images generated by the two systems are randomly shuffled to avoid any annotation bias. Evaluators were asked two simple binary (yes:1/no:0) questions about each system. The questions evaluated: (Q1) does the system perform the desired change mentioned in the input instruction, (Q2) does the system introduces any undesired changes other than the required ones. See Appendix Table 20 for the exact text of the questions. There were a total of 7 evaluators, and each was given the same set of 30 random image quadruples. Table 4 shows the average scores of evaluators across different questions. NEUROSIM performs much better on Q1 despite TIM-GAN using full annotation data, implying better semantic manipulation by NEUROSIM. TIM-GAN does significantly better on Q2 demonstrates its ability to generate better images. The average Fleiss' kappa score (Fleiss et al., 2013) is 0.796, implying high inter-evaluator agreement.

| Qn. | NEUROSIM 5.4K | TIM-GAN 54K |
|-----|---------------|-------------|
| Q1  | 0.43          | 0.31        |
| Q2  | 0.14          | 0.77        |

Table 4: Average human evaluation scores.

| Method | $R1$ | $R3$ |
|--------|------|------|
| GeNeVA $54K$ | 5.0 | 65.8 |
| GeNeVA $5.4K$ | 8.2 | 64.6 |
| TIM-GAN $54K$ | 66.3 | 92.4 |
| TIM-GAN $5.4K$ | 30.2 | 80.7 |
| NEUROSIM $5.4K$ | 63.7 | 89.1 |

Table 5: Performance on generalization to Larger Scenes

| Method | $R1$ | $R3$ |
|--------|------|------|
| Text-Only | 0.2 | 0.4 |
| Image-Only | 34.1 | 83.6 |
| Concat | 39.5 | 86.9 |
| TIRG | 34.8 | 84.6 |
| NEUROSIM | 85.8 | 92.9 |

Table 6: Quality assessment of $G_{\widetilde{I}}$ via image retrieval task.

### 4.6 QUANTITATIVE ASSESSMENT OF MANIPULATED SCENE GRAPH $G_{\widetilde{I}}$

We strongly believe *image rendering module* of NEUROSIM pipeline and *encoder* modules used for computing Recall@$k$ adds some amount of inefficiencies resulting in lower $R1$ and $R3$ scores for us. Therefore, we decide to assess the quality of manipulated scene graph $G_{\widetilde{I}}$ that gets generated in our pipeline. For this, we consider the task of *text guided image retrieval* as proposed by Vo et al. (2019). In this task, an image from the database has to retrieved which would be the closest match to the desired manipulated image but no manipulated image needs to be generated. Therefore, we use our manipulated scene graph $G_{\widetilde{I}}$ as the latent representation of the input instruction and image for image retrieval. We retrieve images from the database based on a novel *graph edit distance* between NEUROSIM generated $G_{\widetilde{I}}$ of the desired manipulated images, and scene graphs of the images in the database. This distance is defined using the *Hungarian algorithm* (Kuhn, 1955) with a simple cost defined between any 2 nodes of the graph. See Appendix D for a detailed explanation. Table 6 captures the performance of NEUROSIM and other popular baselines for the image retrieval task. From this table, we observe that NEUROSIM significantly outperforms supervised learning baselines by a margin of about $50\%$ without ever using output image supervision. This result demonstrates that NEUROSIM edits the scene graph in a meaningful way.

Refer to Appendix D, K, for additional results and ablations respectively.

### 5 CONCLUSION

We present an neuro-symbolic, interpretable approach NEUROSIM to solve image manipulation task using weak supervision of VQA annotations, building on existing work on neuro-symbolic VQA (Mao et al., 2019). Unlike previous approaches, ours is the first work that can handle multi-object scenes with complex instructions requiring multi-hop reasoning, and solve the task without any output image supervision. Our experiments on a newly created dataset of image manipulation demonstrates the potential of our approach compared to supervised baselines. Directions for future work include carefully understanding the nature of errors made by our symbolic programs, and have a human in the loop to provide feedback to the system for correction. Another direction would be experimenting with more complex and real image datasets; recent works on Neuro-symbolic VQA for real images Li et al. (2019) can be a good starting point.

## 6 ETHICS STATEMENT

All the datasets used in this paper were synthetically generated and do not contain any personally identifiable information or offensive content. The ideas and techniques proposed in this paper are useful in designing interpretable natural language-guided tools for image editing, computer-aided design, and video games. One of the possible adverse impacts of AI-based image manipulation is the creation of *deepfakes* Vaccari & Chadwick (2020) (using deep learning to create fake images). To counter deepfakes, several researchers Dolhansky et al. (2020); Mirsky & Lee (2021) have also looked into the problem of detecting real vs. fake images.

## 7 REPRODUCIBILITY STATEMENT

Code for baselines in all our experiments are publicly available, as stated in Section 4. All the training details (e.g., data splits, data processing steps, hyperparameters) are provided in Section 4, Appendix B, and Appendix G. We use the CLEVR dataset (Johnson et al., 2017b) and CLEVR toolkit (code to generate the dataset) for creating the new datasets introduced in this work. These are publicly available to use. Data creation methodology has been explained in Appendix B. Code for NEUROSIM will be open-sourced post acceptance.

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

APPENDIX

## A    DOMAIN SPECIFIC LANGUAGE (DSL)

Table 7 captures the DSL used by our NEUROSIM pipeline. The first 5 constructs in this table are common with the DSL used in Mao et al. (2019). The last 3 operations (`Change,` `Add,` and `Remove`) were added by us to allow for the manipulation operations. Table 8 show the type system

| Operation | Signature [*Output ← Input*]) | Semantics |
|---|---|---|
| Scene | ObjSet ← () | Returns all objects in the scene. |
| Filter | ObjSet ← (ObjSet, ObjConcept) | Filter out a set of objects from ObjSet that have concept (e.g. red) specified in ObjConcept. |
| Relate | ObjSet ← (ObjSet, RelConcept, Obj) | Filter out a set of objects from ObjSet that have concept specified relation concept (e.g. RightOf) with object Obj. |
| Query | ObjConcept ← (Obj, Attribute) | Returns the Attribute value for the object Obj. |
| Exist | Bool ← (ObjSet) | Checks if the set ObjSet is empty. |
| Change | Obj ← (Obj, Concept) | Changes the attribute value of the input object (Obj), corresponding to the input concept, to Concept |
| Add | Graph ← (Graph, RelConcept, Obj, ConceptSet) | Adds an object to the input graph, generating a new graph having the object with attribute values as ConceptSet, and present in relation RelConcept of the input Obj |
| Remove | Graph ← (Graph, ObjSet) | Removes the input objects and their edges from the input graph to output a new graph |

Table 7: Extended Domain Specific Language (DSL) used by NEUROSIM.

used by the DSL in this work. The first 5 types are inherited from Mao et al. (2019) while the last one is an extension of the type system for handling the inputs to the `Add` operator.

| Type | Remarks |
|---|---|
| ObjConcept | Concepts for any given object, such as *blue, cylinder,* etc. |
| Attribute | Attributes for any given object, such as *color, shape,* etc. |
| RelConcept | Relational concepts for any given object pair, such as *RightOf, LeftOf,* etc. |
| Object | Depicts a single object |
| ObjectSet | Depicts multiple objects |
| ConceptSet | A set of elements of ObjConcept type |

Table 8: Extended type system for the DSL used by NEUROSIM.

# B   DATASET DETAILS

## B.1   CIM-NLI DATASET

This dataset was generated with the help of CLEVR toolkit (Johnson et al., 2017b) by using following recipe.

1. First, we create a source image $I$ and the corresponding scene data by using *Blender* (Community, 2018) software.

2. For each source image $I$ created above, we generate multiple instruction texts $T$'s using its scene data. These are generated using templates, similar to question templates proposed by Johnson et al. (2017b).

3. For each such $(I, T)$ pair, we attach a corresponding symbolic program $P$ (not used by NEU-ROSIM though) as well as scene data for the corresponding changed image.

4. Finally, for each $(I, T)$ pair, we generate the target gold image $\widetilde{I}^*$ using Blender software and its scene data from previous step.

Below are some of the important characteristics of the CIM-NLI dataset.

- Each source image $I$ comprises several objects and each object comprises four visual attributes - *color, shape, size,* and *material*.

- Each instructions text $T$ comprises one of the following three kinds of manipulation operations - *add, remove,* and *change*.

- An *add* instruction specifies *color, shape, size,* and *material* of the object that needs to be added. It also specifies a direct (or indirect) relation with one or more existing objects (called reference object(s)). The number of relations that are required to traverse for nailing down the target object is referred to as *# of reasoning hops* and we have allowed instructions with up to 3-*hops reasoning*. We do not generate any 0-hop instruction for *add* due to ambiguity of where to place the object inside the scene.

- A *change* instruction first specifies zero or more attributes to uniquely identify the object that needs to be changed. It may also specify a direct (or indirect) relation with one or more existing reference objects. Lastly, it specifies the target values of an attribute for the identified object which needs to be changed.

- A *remove* instruction specifies zero or more attributes of the object(s) to be removed. Additionally, it may specify a direct (or indirect) relation with one or more existing reference objects.

Table 9 captures the fine grained statistics about the CIM-NLI dataset. Specifically, it further splits each of the *train, validation,* and *test* set across the instruction types - *add, remove,* and *change*.

## B.2   CIM-NLI-LARGE DATASET

We created another dataset called CIM-NLI-LARGE to test the generalization ability of NEUROSIM on images containing more number of objects than training images. CIM-NLI-LARGE tests the *zero-shot transfer* ability of both NEUROSIM and baselines on scenes containing more objects.

Each image in CIM-NLI-LARGE dataset comprises of $10 - 13$ objects as opposed to $3 - 8$ objects in CIM-NLI dataset which was used to train NEUROSIM. The CIM-NLI-LARGE dataset consists of $1K$ unique input images. We have created 3 instructions for each image resulting in a total of $3K$ instructions. The number of *add* instructions is significantly less since there is very little free space available in the scene to add new objects. To create scenes with 12 and 13 objects, we made all objects as *small size* and the minimum distance between objects was reduced so that all objects could fit in the scene. Table 10 captures the statistics about this dataset.

## B.3   MULTI-HOP INSTRUCTIONS

In what follows, we have given examples of the instructions that require multi-hop reasoning to nail down the location/object to be manipulated in the image.

| Operation | Split | # $(I, T, \widetilde{I}^*)$ | # reasoning hops | | | # objects | | |
|---|---|---|---|---|---|---|---|---|
| | | | min | mean | max | min | mean | max |
| Add | train | 17827 | 1 | 2.00 | 3 | 3 | 5.51 | 8 |
| | valid | 4459 | 1 | 2.00 | 3 | 3 | 5.50 | 8 |
| | test | 4464 | 1 | 2.00 | 3 | 3 | 5.45 | 8 |
| Remove | train | 15999 | 0 | 1.50 | 3 | 3 | 5.50 | 8 |
| | valid | 5000 | 0 | 1.50 | 3 | 3 | 5.50 | 8 |
| | test | 5000 | 0 | 1.50 | 3 | 3 | 5.48 | 8 |
| Change | train | 19990 | 0 | 1.50 | 3 | 3 | 5.45 | 8 |
| | valid | 4996 | 0 | 1.50 | 3 | 3 | 5.56 | 8 |
| | test | 4998 | 0 | 1.50 | 3 | 3 | 5.52 | 8 |

Table 9: Statistics of CIM-NLI dataset introduced in this paper.

| Operation | # $(I, T, \widetilde{I}^*)$ | # reasoning hops | | | # objects | | |
|---|---|---|---|---|---|---|---|
| | | min | mean | max | min | mean | max |
| Add | 393 | 1 | 2.0 | 3 | 10 | 11.53 | 13 |
| Remove | 524 | 0 | 1.50 | 3 | 10 | 11.48 | 13 |
| Change | 2083 | 0 | 1.51 | 3 | 10 | 11.50 | 13 |

Table 10: Statistics of CIM-NLI-LARGE dataset.

- *Remove the tiny green rubber ball.* (0-hop)

- *There is a block right of the tiny green rubber ball, remove it.* (1-hop)

- *Remove the shiny cube left of the block in front of the gray thing.* (2-hop)

- *Remove the small thing that is left of the brown matte object behind the tiny cylinder that is behind the big yellow metal block.* (3-hop)

## C  MODEL DETAILS

### C.1  SEMANTIC PARSER

#### C.1.1  DETAILS ON PARSING

We begin by extending the type system of Mao et al. (2019) and add `ConceptSet` because our *add* operation takes as input a set of concepts depicting attribute values of the new object being added (refer Table 8 for the details). Next, in a manner similar to Mao et al. (2019), we use a rule based system for extracting concept words from the input text. We, however, add an extra rule for extracting `ConceptSet` from the input sentence. Rest of the semantic parsing methodology remains the same as given in Mao et al. (2019), with the difference being that our training is weakly supervised (refer Section 3.3 of the main paper).

### C.1.2 TRAINING

As explained in Section 3.3 of the main paper, for training with weaker form of supervision, we use an off-policy program search based REINFORCE (Williams, 1992) algorithm for calculating the exact gradient. For this, we define a set of all possible program templates $\mathbb{P}_t$. For a given input instruction text $T$, we create a set of all possible programs $\{P_T\}$ from $\mathbb{P}_t$. For e.g. given a template $\{remove(relate(\cdot, filter(\cdot, scene())))\}$, this is filled in all possible ways, with concepts, conceptSet, attributes and relational concepts extracted from the input sentence to get programs for this particular template. All such programs created using all templates form the set $P_T$. All $P_T$ are executed over the scene graph of the input image. A typical program structure in our work is of the form *manip_op(reasoning())*, where *manip_op* represents the manipulation operator, for example *change, add,* or *remove*; and *reasoning()* either selects objects for *change* or *remove*, or it selects a reference object for adding another object in relation to it. After a hyperparameter search for the reward (refer Section G of the appendix), we assign a reward of +8 if the *reasoning()* part of the program leads to an object being selected for *change/remove* instruction or a related object being selected for *add* instruction. If no such object is selected, we give a reward of +2. Reward values were decided on the basis of validation set accuracy. We find that with this training strategy, we achieve the validation set accuracy of $95.64\%$, where this accuracy is calculated based on whether a program lead to an object being selected or not. Note, this is a proxy to the actual accuracy. For finding the actual accuracy, we would need a validation set of (instruction, ground truth output program) pairs, but we do not use this supervised data for training or validation.

### C.2 MANIPULATION NETWORK

In what follows, we provide finer details of manipulation network components.

**Change Network:**  As described in Section 3.3 of the main paper, we have a *change neural network* for each attribute. For changing the current attribute value of a given object $o$, we use the following neural network: $\widetilde{o} = g_a(o; c_{s_a^*})$, where $s_a^*$ is the desired changed value for the attribute $a$. $\widetilde{o}$ is the new representation of the object. We model $g_a(\cdot)$ by a single layer neural network without having any non-linearity. The input dimension of this neural network is $(256 + 64)$ because we concatenate the object representation $o \in \mathbb{R}^{256}$ with the desired concept representation $d \in \mathbb{R}^{64}$. We pass this concatenated vector through $g_a(\cdot)$ to get the revised representation of the object: $\widetilde{o} \in \mathbb{R}^{256}$.

The loss used to train the weights of the change network is a weighted sum of losses equation 1 to equation 4 given in the main paper. This leads to the overall loss function given below.

$$L_{\text{overall\_change}} \quad = \quad \lambda_1\, \ell_a + \lambda_2\, \ell_{\overline{a}} + \lambda_3\, \ell_{\text{cycle}} + \lambda_4\, \ell_{\text{consistency}} + \lambda_5\, \ell_{\text{objGAN}} \tag{8}$$

where, $\ell_{\text{objGAN}}$ above is the modified GAN loss (Goodfellow et al., 2014). Here $\lambda_1 = 1$, $\lambda_2 = 1/((\text{num\_attrs} - 1) * (\text{num\_concepts}))$, $\lambda_3 = \lambda_4 = 10^3$, and $\lambda_5 = 1/(\text{num\_objects})$. Here, (num\_objects) is the number of objects in input image, (num\_attrs) is the total number of attributes for each object, and (num\_concepts) are the total number of concepts in the NSCL (Mao et al., 2019) framework.

The object discriminator is a neural network with input dimension 256 and a single 300 dimensional hidden layer with ReLU activation function. This discriminator is trained using standard GAN objective $\ell_{\text{objGAN}}$. See Fig 5a for an overview of the change operator

**Remove Network:**  The remove network is a symbolic operation as described in Section 3.3 of the main paper. That is, given an input set of objects, the remove operation deletes the subgraph of the scene graph that contains the nodes corresponding to removed objects and the edges incident on those nodes. See Fig 5c for an overview of the remove operator.

**Add Network:**  The neural operation in the add operator comprises of predicting the object representation for the newly added object using a function $g_{\text{addObj}}(\cdot)$. This function is modeled as a single layer neural network without any activation. The input to this network is a concatenated vector $[[c_{s_{a_1}}, c_{s_{a_2}}, \cdots, c_{s_{a_k}}], o_{rel}, c_r]$, where $[c_{s_{a_1}}, c_{s_{a_2}}, \cdots, c_{s_{a_k}}]$ represents the concatenation of all the concept vectors of the desired new objects. The vector $o_{rel}$ is the representation of the object with whom the relation (i.e. position) of the new object has been specified and $c_r$ is the concept vector for

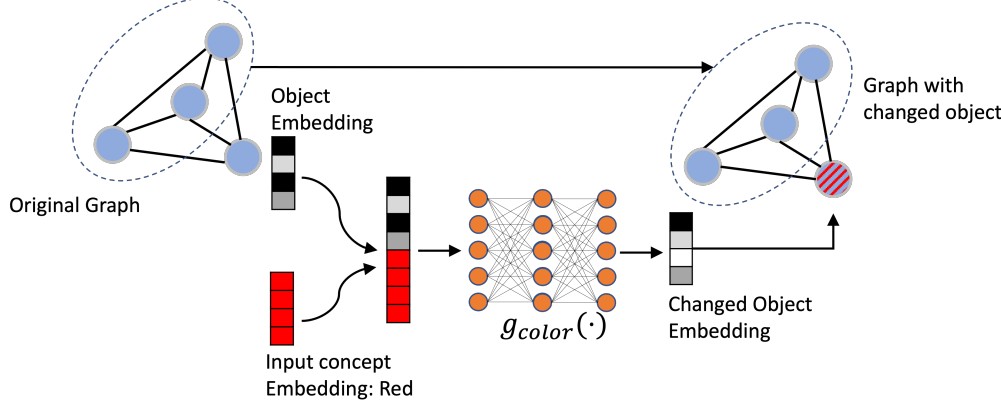

(a) Change operator overview.

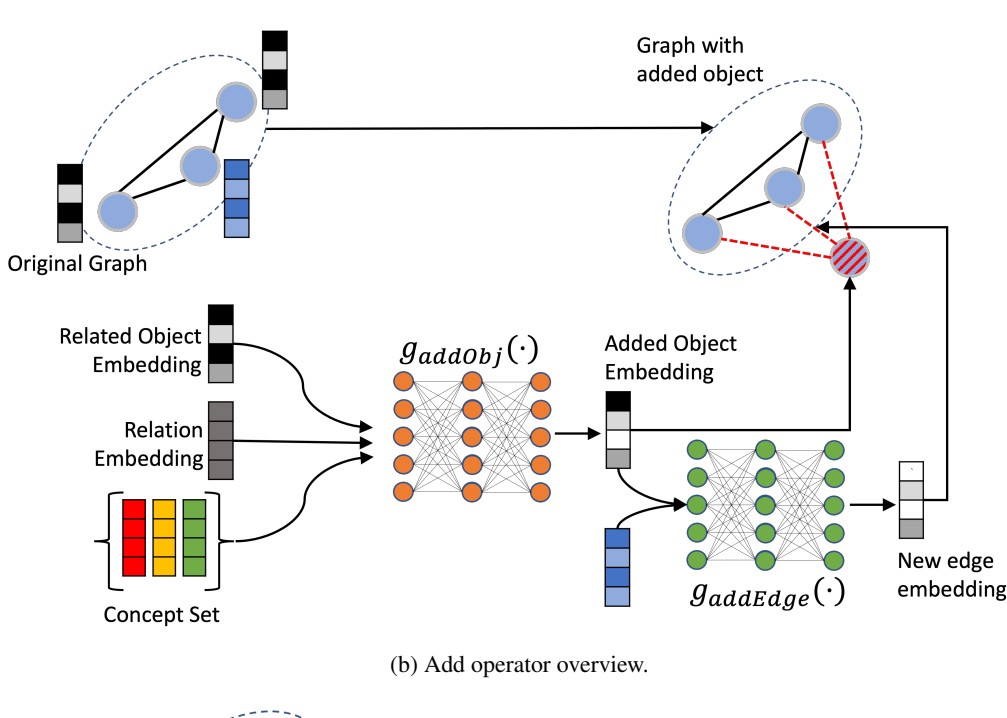

(b) Add operator overview.

(c) Remove operator overview.

Figure 5: Overview of new operators (*change, add* and *remove*) added to the DSL.

that relationship. The input dimension of $g_{\text{addObj}}(\cdot)$ is $(k * 64 + 256 + 64)$ and the output dimension is 256. For predicting representation of newly added edges in the scene graph, we us edge predictor $g_{\text{addEdge}}(\cdot)$. The input to this edge predictor function is the concatenated representation of the objects

which are linked by the edge. The input dimension of $g_{\text{addEdge}}(\cdot)$ is $(256 + 256)$ and the output dimension is 256.

The loss used to train the *add network* weights is a weighted sum of losses equation 5 to equation 7 along with an object discriminator loss. The overall loss is given by the following expression.

$$
\begin{aligned}
L_{\text{overall\_add}} \quad = \quad & \lambda_1 \ell_{\text{concepts}} + \lambda_2 \ell_{\text{relation}} + \lambda_3 \ell_{\text{objSup}} + \lambda_4 \ell_{\text{edgeSup}} + \\
& \lambda_5 \ell_{\text{edgeGAN}} + \lambda_6 \ell_{\text{objGAN}}
\end{aligned}
\quad (9)
$$

where, $\ell_{\text{objGAN}}$ and $\ell_{\text{edgeGAN}}$ above denotes the modified GAN loss (Goodfellow et al., 2014). Here $\lambda_1 = \lambda_2 = 1/(\text{num\_attrs})$, $\lambda_3 = \lambda_4 = 10^3$, $\lambda_5 = 1/(\text{num\_objects})$.

The object discriminator is a neural network with input dimension as 256 and a single 300 dimensional hidden layer with ReLU activation function. This discriminator is trained using the standard GAN objective $\ell_{\text{objGAN}}$. Note, $\ell_{\text{objGAN}}$ has 2 parts – i) the loss for the generated (fake) object embedding using the *add network*, and ii) the loss for the real objects (all the unchanged object embeddings of the image). The former is unscaled but the latter one is scaled by a factor of $1/(\text{num\_objects})$.

The edge discriminator is a neural network with input dimension as $(256 * 3)$ and a single 300 dimensional hidden layer with ReLU activation function. As input to this discriminator network, we pass the concatenation of the two objects and the edge connecting them. This discriminator is trained using the standard GAN objective $\ell_{\text{edgeGAN}}$. See Fig 5b for an overview of the add operator

## D  ADDITIONAL RESULTS

### D.1  DETAILED PERFORMANCE FOR ZERO-SHOT GENERALIZATION ON LARGER SCENES

Table 11 below is a detailed version of the table 5 in the main paper. This table compares the performance of NEUROSIM with baseline methods TIM-GAN and GeNeVA for the zero-shot generalization to larger scenes (with $\geq 10$ objects), while the models were trained on images with $3 - 8$ objects. Relative to the main paper's table 5, this table offers separate performance numbers for each of the *add, remove* and *change* instructions.

| Method | Train Data Size | Add | | Change | | Remove | |
|---|---|---|---|---|---|---|---|
| | | $R1$ | $R3$ | $R1$ | $R3$ | $R1$ | $R3$ |
| GeNeVA | $54K$ | 0.5 | 64.6 | 4.9 | 69.9 | 9.0 | 50.0 |
| GeNeVA | $5.4K$ | 0.0 | 60.1 | 8.2 | 69.2 | 14.3 | 49.6 |
| TIMGAN | $54K$ | 12.5 | 77.4 | 73.4 | 95.2 | 78.2 | 92.2 |
| TIMGAN | $5.4K$ | 1.0 | 70.0 | 32.1 | 84.4 | 44.7 | 74.0 |
| NEUROSIM | $5.4K$ | 3.8 | 46.6 | 68.2 | 95.8 | 90.7 | 94.3 |

Table 11: Detailed performance scores for NEUROSIM, TIM-GAN, and GeNeVA for zero-shot generalization to larger scenes (with $\geq 10$ objects) from CIM-NLI-LARGE dataset, while models are trained on images with $3 - 8$ objects. Table has separate performance numbers for *add, remove,* and *change* instructions. Along with each method, we have also written the number of data points from CIM-NLI dataset that were used for training. $R1$ and $R3$ correspond to Recall@1 and Recall@3, respectively.

### D.2  IMAGE RETRIEVAL TASK

A task that is closely related to the image manipulation task is the task of *Text Guided Image Retrieval*, proposed by Vo et al. (2019). Through this experiment, our is to demonstrate that NEUROSIM is highly effective in solving this task as well. In what follows, we provide details about this task, baselines, evaluation metric, how we adapted NEUROSIM for this task, and finally performance results in Table 12. This table is a detailed version of the Table 6 in the main paper.

**Task Definition:**    Given an Image $I$, a text instruction $T$, and a database of images $D$, the task is to retrieve an image from the database that is semantically as close to the ground truth manipulated image as possible.

Note, for each such $(I, T)$ pair, some image from the database, say $\widetilde{I} \in D$, is assumed to be the ideal image that should ideally be retrieved at rank-1. This, so called desired gold retrieval image might even be an image which is the ideal manipulated version of the original images $I$ in terms of satisfying the instruction $T$ perfectly. Or, image $\widetilde{I}$ may not be such an ideal manipulated image but it still may be the image in whole corpus $D$ that comes closest to the ideal manipulated image.

In practice, while measuring the performance of any such system for this task, the gold manipulated image for $(I, T)$ pair is typically inserted into the database $D$ and such an image then serves as the desired gold retrieval image $\widetilde{I}$.

**Baselines:**    Our baselines includes popular supervised learning systems designed for this task. The first baseline is TIRG proposed by Vo et al. (2019) where they combine image and text to get a joint embedding and train their model in a *supervised* manner using embedding of the desired retrieved image as supervision. For completeness, we also include comparison with other baselines – *Concat, Image-Only*, and *Text-Only* – that were introduced by Vo et al. (2019).

A recent model proposed by Chen et al. (2020) uses symbolic scene graphs (instead of embeddings) to retrieve images from the database. Motivated by this, we also retrieve image via the scene graph that is generated by the manipulation module of NEUROSIM. However, unlike Chen et al. (2020), the nodes and edges in our scene graph have associated vectors and make a novel use of them while retrieving. We do not compare our performance with Chen et al. (2020) since it's code is unavailable and we haven't been able to reproduce their numbers on datasets used in their paper. Moreover, Chen et al. (2020) uses full supervision of the desired output image (which is converted to a symbolic scene graph), while we do not.

**Evaluation Metric:**    We use Recall@$k$ (and report results for $k = 1, 3$) for evaluating the performance of text guided image retrieval algorithms which is standard in the literature.

**Retrieval using Scene Graphs:**    We use the scene graph generated by NEUROSIM as the latent representation to retrieve images from the database. We introduce a novel yet simple method to retrieve images using scene graph representation. For converting an image into the scene graph, we use the visual representation network of NEUROSIM. Given the scene graph $G$ for the input image $I$ and the manipulation instruction text $T$, NEUROSIM converts the scene graph into the changed scene graph $G_{\widetilde{I}}$, as described in Section C in Appendix. Now, we use this graph $G_{\widetilde{I}}$ as a query to retrieve images from the database $D$. For retrieval, we use the novel graph edit distance (GED) between $G_{\widetilde{I}}$ and the scene graph representation of the database images. The scene graph for each database image is also obtained using the visual representation network of NEUROSIM. The graph edit distance is given below.

$$GED(G_{\widetilde{I}}, G_D) \quad = \quad \begin{cases} \infty & |N_{\widetilde{I}}| \neq |N_{\widetilde{D}}| \\ \min_{\pi \in \Pi} \sum_{\forall i \in \{1, 2, \cdots, |N_{\widetilde{I}}|\}} c(n_i, y_i) & \text{otherwise.} \end{cases}$$

where, $G_{\widetilde{I}} = (N_{\widetilde{I}}, V_{\widetilde{I}})$ and $G_D = (N_D, V_D)$. $n_i$ and $y_i$ are the node embeddings of the query graph $G_{\widetilde{I}}$ and scene graph $G_D$ of an image from the database. $c(a, b)$ is the cosine similarities between embeddings $a$ and $b$. This GED is much simpler than that defined in Chen et al. (2020), since it does not need any hand designed cost for *change, removal,* or *addition* of nodes, or different attributes values. It can simply rely on the cosine similarities between node embeddings. We use the *Hungarian algorithm* (Kuhn, 1955) for calculating the optimal matching $\pi$ of the nodes, among all possible matching $\Pi$. We use the negative of the cosine similarity scores between nodes to create the cost matrix for the Hungarian algorithm to process. This simple yet highly effective approach (See Table 6 in the main paper and Table 12 in the appendix), can be improved by more sophisticated techniques that include distance between edge embeddings and including notion of subgraphs in the GED. We leave this as future work. This result shows that our manipulation network edits the scene graph in a desirable manner, as per the input instruction.

| Method | Train Data Size | Add | | | Change | | | | Remove | | | |
|--------|------|------|------|------|------|------|------|------|------|------|------|------|
| | | 1 | 2 | 3 | 0 | 1 | 2 | 3 | 0 | 1 | 2 | 3 |
| Text-Only | 54$K$ | 0.4 | 0.3 | 0.3 | 0.1 | 0.0 | 0.1 | 0.1 | 0.1 | 0.3 | 0.3 | 0.0 |
| Image-Only | 54$K$ | 35.3 | 33.4 | 32.3 | 20.1 | 23.2 | 16.9 | 19.8 | 46.3 | 41.3 | 53.1 | 57.8 |
| Concat | 54$K$ | 36.3 | 33.3 | 31.8 | 37.3 | 40.4 | 34.2 | 37.9 | 41.8 | 41.0 | 50.0 | 55.0 |
| TIRG | 54$K$ | 35.6 | 31.8 | 33.5 | 22.0 | 25.1 | 18.8 | 22.0 | 46.6 | 42.7 | 52.5 | 56.1 |
| NEUROSIM | 5.4$K$ | 96.2 | 95.3 | 95.3 | 83.3 | 82.9 | 81.3 | 78.7 | 79.6 | 77.4 | 86.4 | 82.2 |

Table 12: Performance scores (Recall@1) on the Image Retrieval task, comparing NEUROSIM with TIM-GAN and GeNeVA with increase in reasoning hops, for *add, remove,* and *change* instructions. Along with each method, number of data points from CIM-NLI used for training are written.

| Method | Instruction | $\beta = 0.054$ | | $\beta = 0.07$ | | $\beta = 0.1$ | | $\beta = 0.2$ | | $\beta = 0.54$ | |
|--------|-------------|------|------|------|------|------|------|------|------|------|------|
| | | $R1$ | $R3$ | $R1$ | $R3$ | $R1$ | $R3$ | $R1$ | $R3$ | $R1$ | $R3$ |
| GeNeVA | add | 0.0 | 57.3 | – | – | – | – | – | – | 0.7 | 63.6 |
| | change | 5.9 | 36.3 | – | – | – | – | – | – | 4.1 | 39.4 |
| | remove | 13.2 | 82.3 | – | – | – | – | – | – | 8.7 | 89.3 |
| TIM-GAN | add | 1.9 | 70.7 | 4.9 | 74.0 | 8.6 | 76.7 | 10.3 | 77.1 | 13.1 | 78.6 |
| | change | 41.0 | 72.1 | 42.9 | 73.5 | 49.8 | 77.3 | 62.5 | 84.2 | 78.3 | 92.3 |
| | remove | 49.6 | 79.5 | 47.0 | 91.9 | 53.9 | 93.1 | 65.3 | 96.8 | 78.0 | 98.5 |
| NEUROSIM | add | 4.9 | 30.9 | 6.4 | 34.8 | 5.7 | 34.7 | 5.9 | 38.9 | 5.6 | 35.0 |
| | change | 57.2 | 79.4 | 57.3 | 79.3 | 57.2 | 79.3 | 57.2 | 79.4 | 57.1 | 79.3 |
| | remove | 69.6 | 82.5 | 69.5 | 82.5 | 69.5 | 82.6 | 69.5 | 82.5 | 69.6 | 82.5 |

Table 13: Detailed performance comparison of NEUROSIM with TIM-GAN (Zhang et al., 2021) and GeNeVA (El-Nouby et al., 2019) with varying $\beta$ levels, split across add, remove and change instructions. The '-' entries for GeNeVA were not computed due to excessive training time; it's performance is abysmal even when using full data. We always use 100$K$ VQA examples (5K Images, 20 questions per image) for our weakly supervised training. $R1$ and $R3$ correspond to Recall@1 and 3, respectively. For Recall, higher the score is better.

## D.3 DETAILED MULTI-HOP REASONING PERFORMANCE

Table 14 below provides a detailed split of the performance numbers reported in Table 3 of the main paper across i) number of hops ($0 - 3$ hops) and ii) type of instructions *(add/remove/change)*. We observe that for *change* and *remove* instructions, NEUROSIM improves over TIM-GAN and GeNeVA trained on $5.4K$ CIM-NLI data points by a significant margin ($\sim 20\%$ on 3-hop *change/remove* instructions). However, NEUROSIM lags behind TIM-GAN when the entire CIM-NLI labelled data is used to train TIM-GAN. We also observe that all the models perform poorly on the *add* instructions, as compared to *change* and *remove* instructions.

## D.4 DETAILED PERFORMANCE FOR DIFFERENT COST RATIOS $\beta$

Table 2 in Section 4 of the main paper showed the performance of NEUROSIM compared with TIM-GAN and GeNeVA for various values of $\beta$, where $\beta$ is the ratio of the number of annotated

| Method | Train Data Size | Add | | | Change | | | | Remove | | | |
|--------|-----------------|-----|-----|-----|--------|-----|-----|-----|--------|-----|-----|-----|
| | | 1 | 2 | 3 | 0 | 1 | 2 | 3 | 0 | 1 | 2 | 3 |
| GeNeVA | 54$K$ | 1.1 | 0.5 | 0.5 | 3.6 | 4.2 | 4.7 | 3.9 | 9.0 | 8.0 | 8.3 | 9.4 |
| GeNeVA | 5.4$K$ | 0.0 | 0.0 | 0.0 | 4.7 | 7.1 | 5.6 | 6.1 | 12.3 | 11.3 | 15.5 | 13.5 |
| TIM-GAN | 54$K$ | 7.6 | 16.1 | 15.7 | 85.8 | 74.1 | 78.0 | 75.4 | 82.2 | 68.3 | 81.9 | 79.7 |
| TIM-GAN | 5.4$K$ | 1.4 | 2.3 | 1.9 | 54.5 | 36.4 | 38.7 | 34.5 | 58.3 | 40.9 | 50.9 | 48.2 |
| NEUROSIM | 5.4$K$ | 4.6 | 5.0 | 5.1 | 59.5 | 57.9 | 55.8 | 55.7 | 69.6 | 66.6 | 71.8 | 70.4 |

Table 14: Performance scores (Recall@1) for NEUROSIM with TIM-GAN and GeNeVA with increase in reasoning hops, for *add, remove,* and *change* instructions. Along with each method, number of data points from CIM-NLI used for training are written.

(with output image supervision) image manipulation examples required by the supervised baselines, to the number of annotated VQA examples required to train NEUROSIM. In Table 13, we show a detailed split of the performance, for the *add, change,* and *remove* operators, across the same values of $\beta$ as taken before.

We find that for the *change* operator, NEUROSIM performs better than TIM-GAN by a margin of $\sim 8\%$ (considering Recall@1) for $\beta \leq 0.1$. For the *remove* operator, NEUROSIM performs better than TIM-GAN by a margin of $\sim 4\%$ (considering Recall@1) for $\beta \leq 0.2$. Overall, NEUROSIM performs similar to TIM-GAN, for $\beta = 0.2$, for *remove* and *change* operators. All models perform poorly on the *add* operator as compared to the *change* and *remove* operators. We find that having full output image supervision allows TIM-GAN to reconstruct (copy) the unchanged objects from the input to the output for all the operators. This results in a higher recall in general but it's effect is most pronounced in the Recall@3. NEUROSIM, on the other hand, suffers from rendering errors which makes the overall recall score (especially Recall@3) lower. We believe that improving image rendering quality would significantly improve the performance of NEUROSIM and we leave this as future work.

## D.5 RESULTS ON DATASETS FROM DIFFERENT DOMAINS

### D.5.1 MINECRAFT DATASET

**Dataset Creation:** We create a new dataset having (Image, instruction) by building over the Minecraft dataset used in Yi et al. (2018). Specifically, we create zero and one hop remove instructions and one hop add instructions similar the creation of CIM-NLI. This dataset contains scenes and objects from the Minecraft video game and is used in prior works for testing Neuro-Symbolic VQA systems like NSCL Mao et al. (2019) and NS-VQA Yi et al. (2018). The setting of the Minecraft worlds dataset is significantly different from CLEVR in terms of concepts and attributes of objects and visual appearance.

**Experiment:** We use the above dataset for testing the addition and removal of objects using NeuroSIM (See Fig 6). We train NeuroSIM's decoder to generate images from scene graphs of the minecraft dataset. We assume access to a parser that gives us programs for an instruction. For removal, we use the same remove network as described above, while for addition, we assume access to the features of object to be added, which is added to the scene graph of the image and the decoder decodes the final image. See Figure 6 for a set of successful examples on the Minecraft dataset. We see that using our method, one can add and remove objects from the scene successfully, without using any output image as supervision during training. Though we have assumed the availability of parser in the above set-up, training it jointly with other modules should be straightforward, and can be achieved using our general approach described in Section 3 of the main paper.

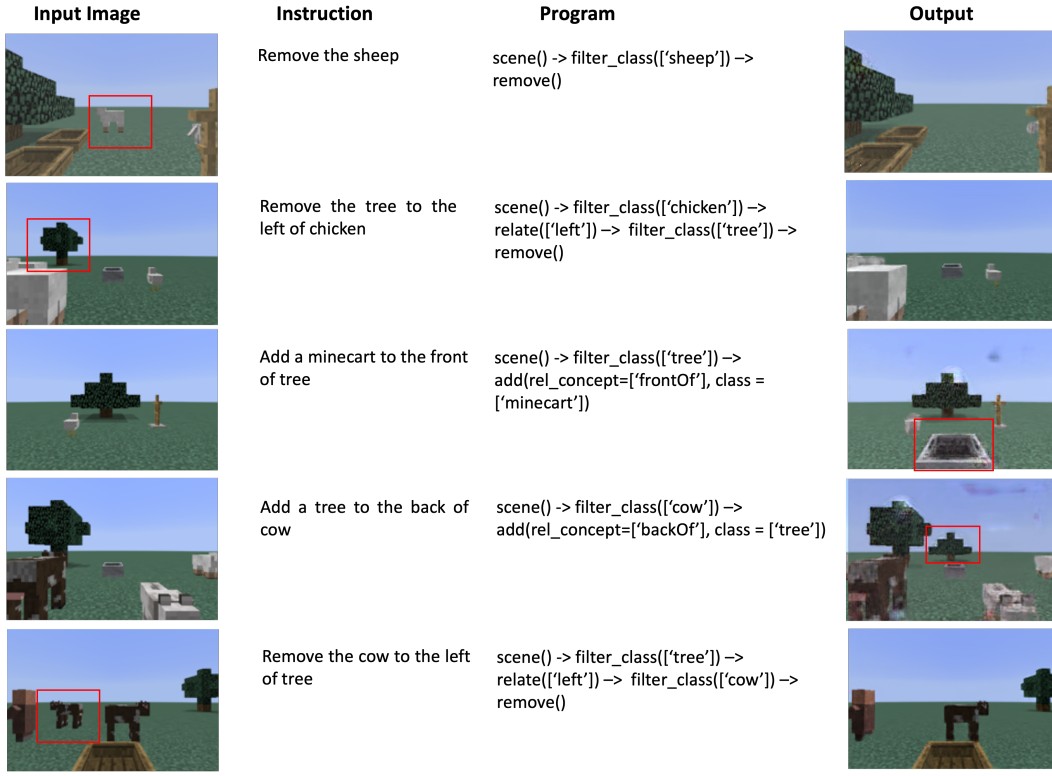

Figure 6: Results for addition and removal of objects from images of the minecraft dataset

### D.5.2 Real World Dataset: GQA

To show that our proposed approach NEUROSIM can work on real-world images, we experimented with the GQA dataset (Hudson & Manning, 2019). This dataset was originally used for benchmarking the task of VQA, and contains real world scenes having multiple objects, with different concepts and attributes.

**1] Zero-Shot Domain Transfer**

Our first experiment was to check the performance of NEUROSIM (trained only on synthetic images from CLEVR dataset) on real-world images without any retraining/fine-tuning. For this, we handpicked a few real images from GQA dataset and performed following steps:

1. Generated a scene graph for the image using our existing visual representation network (without retraining it).
2. Next, we queried for the color of an object in the image using our query network.
3. Next, we changed the color of the given object through our existing manipulation network.
4. Finally, rendered the image using the representation of the changed object.

Figure 7 shows probabilities obtained when we query the representation of real-world objects using our pretrained query networks before and after applying the pretrained change network to these object representations. From this zero-shot experiment, our query network and manipulation networks is able to disentangle attributes such as the object's color and also change it. However, because the rendering module is never trained on real images, it struggles to generate the real images. It seems to map the object with the shapes it learned during CLEVR training. Training the image rendering module using graph-based representations on real images (e.g. object representations obtained using Faster RCNN) is likely to eliminate the above problem and is part of our future and ongoing work.

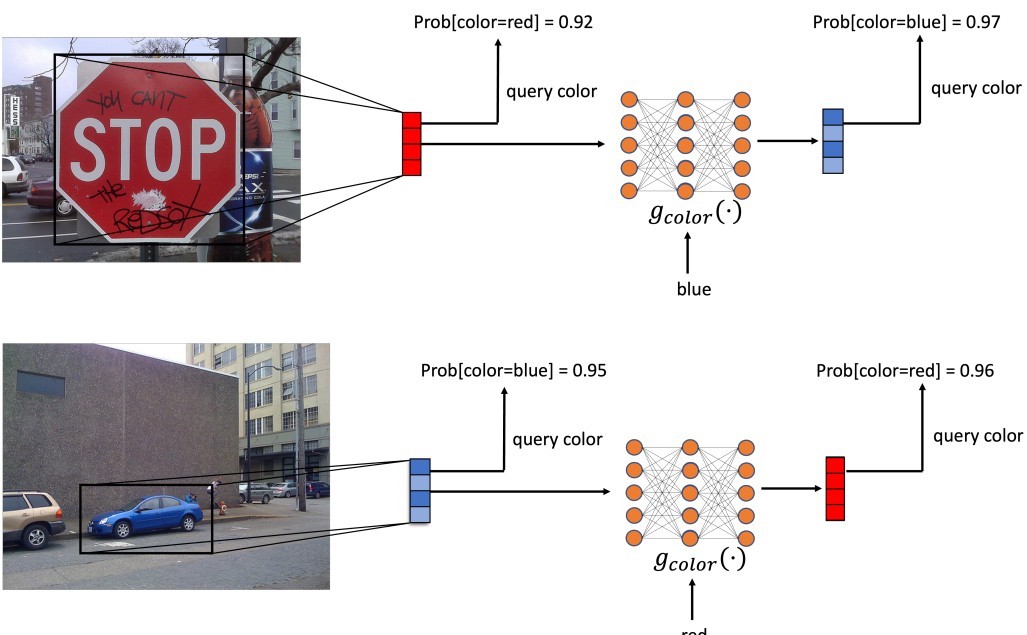

Figure 7: Examples of querying an object representation before and after the application of change operation by NEUROSIM trained only on synthetic images from CLEVR dataset. Example images are taken from GQA dataset Hudson & Manning (2019).

### 2] Training on Real World Dataset

- We trained our *image rendering module* (Section 3.4) from scratch on the GQA dataset. During training, the scene graph of an input image is constructed by extracting image objects' representation vectors by exploiting pre-trained *Faster RCNN* Ren et al. (2015) and *ResNet* He et al. (2016a). This scene graph is fed as input to the rendering module. We want to emphasize that the rendering module is not trained fully to convergence due to a lack of computational resources and time during the discussion phase.

- Next, we performed inference on the above trained *image rendering module* using unseen sample images from the dataset. Following ideas from Mao et al. (2019) and Li et al. (2021) we used a pre-trained *ResNet* classifier as our program parser, for selecting objects in a given scene as well as tagging their class labels (e.g. horse, elephant, etc.) and corresponding probabilities. This is similar to the concept quantization step described in the main paper. For reinforcing the interpretability benefits of our model, we have shown the output of *remove* operation on these examples in Fig 8 as well as the steps taken by the model to achieve this. For each example in this figure,

  - The leftmost image is the source image that needs to be manipulated.
  - The rightmost (bottom) image is obtained after rendering the scene graph of the source image. The purpose of this image is to show the baseline quality of the rendering module.
  - The rightmost (top) image is obtained after manipulation operation (*removal of a single object in this case*) is performed by NEUROSIM.

Fig 8 shows that after applying the *remove* operation to an object in the given image, NEUROSIM is able to reconstruct the image without that object while keeping the rest of the scene intact (when compared with baseline rendered image). We believe a more comprehensive training of the image rendering module to convergence will result in better-quality visuals.

We have demonstrated the *remove* operation on real-world images through the above experiment. We believe this result is still significant, since ours is the the first work in this direction to achieve complex image manipulation through text. Performing the *add* and *change* operations on real-world images is future work.

Figure 8: Examples of the application of remove operation on real images. Please note that the programs are applied on the scene graphs. For better visualization, program steps have been shown on the full image.

### D.5.3 SUMMARY OF RESULTS

We have shown proof-of-concept experiments on two additional datasets: Minecraft (artificial), GQA (real). The goal of our experiments was to demonstrate that our technique has the potential to generalize beyond CLEVR based datasets, including on real images. Doing a more comprehensive set of experiments, such as training the parser on Minecraft along with other modules, as well as, working with change and add instruction on the real dataset, and possibly exploring other powerful decoding techniques (such as based on recent diffusion models Ramesh et al. (2022); Saharia et al. (2022)), is a direction for future work.

## E  END-TO-END TRAINING

The main objective of this work is to make use of weakly supervised VQA data for the image manipulation task without using output image supervision. But a natural extension of our work is to use output image supervision as well, to improve the performance of NEUROSIM. We devised an experiment to compare how much performance boost can be obtained by utilizing ground truth output (manipulated) images as the supervision for different modules of NEUROSIM. This experiment demonstrates the value of end-to-end training for NEUROSIM and how it can exploit the supervised data. We refer to this variant as NEUROSIM(e2e). We begin with a pre-trained NEUROSIM model trained with VQA annotations and then fine-tune it using supervised manipulation data. The detailed results are given in Table 15. This experiment demonstrates that with a small amount of supervised data, the performance of NEUROSIM can be significantly improved (e.g., more than 9 points increase for the change instruction with only $5.4K$ supervision examples)

| Instruction | Model | # of CIM-NLI examples used for training | | | | |
|---|---|---|---|---|---|---|
| | | 5.4K | 7K | 10K | 20K | 54K |
| Change | GeNeVA | 5.9 | - | - | - | 4.1 |
| | TIM-GAN | 41.0 | 42.9 | 49.8 | 62.5 | **78.3** |
| | NEUROSIM | 57.2 | 57.3 | 57.2 | 57.2 | 57.1 |
| | NEUROSIM(e2e) | **66.2** | **66.3** | **66.6** | **67.4** | 69.6 |
| Add | GeNeVA | 0.0 | - | - | - | 0.7 |
| | TIM-GAN | 1.9 | 4.9 | 8.6 | 10.3 | **13.1** |
| | NEUROSIM | 4.9 | 6.4 | 5.7 | 5.9 | 5.6 |
| | NEUROSIM(e2e) | **8.8** | **8.9** | **9.2** | **10.5** | 10.6 |
| Remove | GeNeVA | 13.2 | - | - | - | 8.7 |
| | TIM-GAN | 49.6 | 47.0 | 53.9 | 65.3 | **78** |
| | NEUROSIM | **69.6** | **69.5** | **69.5** | **69.5** | 69.6 |
| | NEUROSIM(e2e) | **69.6** | **69.5** | **69.5** | **69.5** | 69.6 |

Table 15: Performance comparison of NEUROSIM(e2e) with baselines. NEUROSIM(e2e) refers to NEUROSIM trained end-to-end by utilizing ground truth manipulated images as the supervision for NEUROSIM modules.

Given the significant increase in performance of NEUROSIM when using supervised data, we also test it's generalization capability (Analogous to Section 4.2, 4.3), and quality of scene graph retrieval (Analogous to Section 4.5).

From Table 16, we see that NEUROSIM(e2e) shows improved zero-shot generalization to larger scenes. Even when trained on just 5.4k CIM-NLI data, NEUROSIM(e2e) improves over TIM-GAN-54k by 3.9 R@1 points. A 5.3 point improvement over TIM-GAN is observed when full CIM-NLI data is used.

Next, we measure drop in performance with increasing reasoning hops. From Table 17, we see that NEUROSIM(e2e) achieves the lowest drop when compared to TIM-GAN. NEUROSIM(e2e) improves over weakly supervised NEUROSIM baseline by 6.6 R@1 points.

Finally, we measure quality of scene graph via retrieval. From Table 18, we see that supervised training significantly improves the scene graph quality, thus improving retrieval performance. Supervised training improves retrieval by 7.3 R@1 points over weakly supervised NEUROSIM baseline.

| Model | Train Data Size | R1 | R3 |
|---|---|---|---|
| TIM-GAN | 5.4K | 30.2 | 80.7 |
| TIM-GAN | 54K | 66.3 | 92.4 |
| NEUROSIM | 5.4K | 63.7 | 89.1 |
| NEUROSIM(e2e) | 5.4K | 70.2 | **92.6** |
| NEUROSIM(e2e) | 54K | **71.6** | 91.7 |

Table 16: Zero-shot generalization to larger scenes (Extension of Table 5 of main paper).

| Method | Train Data Size | Hops | | Drop in Performance |
|---|---|---|---|---|
| | | $ZH$ | $MH$ | |
| TIM-GAN | 5.4K | 56.4 | 41.6 | -14.8 |
| TIM-GAN | 54K | 84.0 | 76.2 | -7.8 |
| NEUROSIM | 5.4K | 64.5 | 63.0 | **-1.5** |
| NEUROSIM(e2e) | 5.4K | 69.4 | 67.3 | -2.1 |
| NEUROSIM(e2e) | 54K | 71.1 | 69.6 | **-1.5** |

Table 17: Performance with increasing reasoning hops (Extension of Table 3 of main paper).

These findings suggest that NEUROSIM(e2e) significantly outperforms other supervised approaches in almost all settings. One can fine-tune the image decoder and the visual representation network to further enhance the findings, which should greatly enhance the outcomes.

## F    COMPUTATIONAL RESOURCES

We trained all our models and baselines on 1 Nvidia Volta V100 GPU with 32GB memory and 512GB system RAM. Our image decoder training takes about 4 days of training time. Training of the VQA task takes $5 - 7$ days of training time and training the Manipulation networks take $4 - 5$ hours of training time.

| Model | R1 | R3 |
|---|---|---|
| Text-Only | 0.2 | 0.4 |
| Image-Only | 34.1 | 83.6 |
| Concat | 39.5 | 86.9 |
| TIRG | 34.8 | 84.6 |
| NEUROSIM | 85.8 | 92.9 |
| NEUROSIM(e2e) | **93.1** | **96.7** |

Table 18: Quality of scene graph measured via retrieval (Extension of Table 6 of main paper)

## G  HYPERPARAMETERS AND VALIDATION ACCURACIES

### G.1  TRAINING FOR VQA TASK

The hyperparameters for the VQA task are kept same as default values coming from the prior work (Mao et al., 2019). We refer the readers to Mao et al. (2019) for more details. We obtained a question answering accuracy of 99.3% after training on the VQA task.

### G.2  TRAINING SEMANTIC PARSER

The semantic parser is trained to parse instructions. Learning of this module happens using the REINFORCE algorithm as described in Section C of this appendix. During REINFORCE algorithm, we search for positive rewards from the set $\{7, 8, 10\}$, and negative rewards from the set $\{0, 2, 3\}$. We finally choose a positive reward of 8 and negative reward of 2. For making this decision, we first train the semantic parser for 20 epochs and then calculate its accuracy by running it on the quantized scenes from the validation set. For a particular output program, we say it is correct if it leads to an object being selected (see Section C of the appendix for more information) and this is how the accuracy of the semantic parser is calculated. This accuracy is a proxy for the real accuracy. An alternative is to use annotated ground truth programs for calculating accuracy and then selecting hyperparameters. However, we do not use ground truth programs. All other hyperparameters are kept the same as used by Mao et al. (2019) to train the parser on VQA task. We obtain a validation accuracy of 95.64% after training the semantic parser for manipulation instructions.

### G.3  TRAINING MANIPULATION NETWORKS

The architecture details of the manipulation network are present in Section C of this appendix. We use batch size of 32, learning rate of $10^{-3}$, and optimize using AdamW (Loshchilov & Hutter, 2017) with weight decay of $10^{-4}$. Rest of the hyperparameters are kept the same as used in Mao et al. (2019). During training, at every $5^{th}$ epochs, we calculate the manipulation accuracy by using the query networks that were trained while training the NEUROSIM on VQA data. This serves as a proxy to the validation accuracy.

- For the *change* network training, we use the query accuracy of whether the attribute that was suppose to change for a particular object, has changed correctly or not. Also, whether any other attribute has changed or not.

- For the *add* network training, we use the query accuracy of whether the attributes of the added object are correct or not. Also, whether the added object is in a correct relation with reference object or not.

We obtained a validation accuracy (based on querying) of 95.9% for the *add* network and an accuracy of 99.1% for the *change* network.

### G.4  IMAGE DECODER TRAINING

The architecture of the image decoder is similar to Johnson et al. (2018) but our input scene graph (having embeddings for nodes and edges) is directly processed by the graph neural network. We use a batch size of 16, learning rate of $10^{-5}$, and optimize using Adam (Kingma & Ba, 2014) optimizer. The rest of the hyperparameters are same as Johnson et al. (2018). We train the image decoder for a fixed set of $1000K$ iterations.

## H  QUALITATIVE ANALYSIS

Figures 9,10,11 compare the images generated by NEUROSIM, TIM-GAN, and GeNeVA on add, change and remove instructions respectively. NEUROSIM's advantage lies in semantic correctness of manipulated images. For example, see Figure 9 row #3,4; Figure 10 row #2; 11 all images. In these images, NEUROSIM was able to achieve semantically correct changes, while TIM-GAN, GeNeVA faced problems like blurry, smudged objects while adding them to the scene, removing incorrect objects from the scene, or not changing/partially changing the object to be changed. Images generated

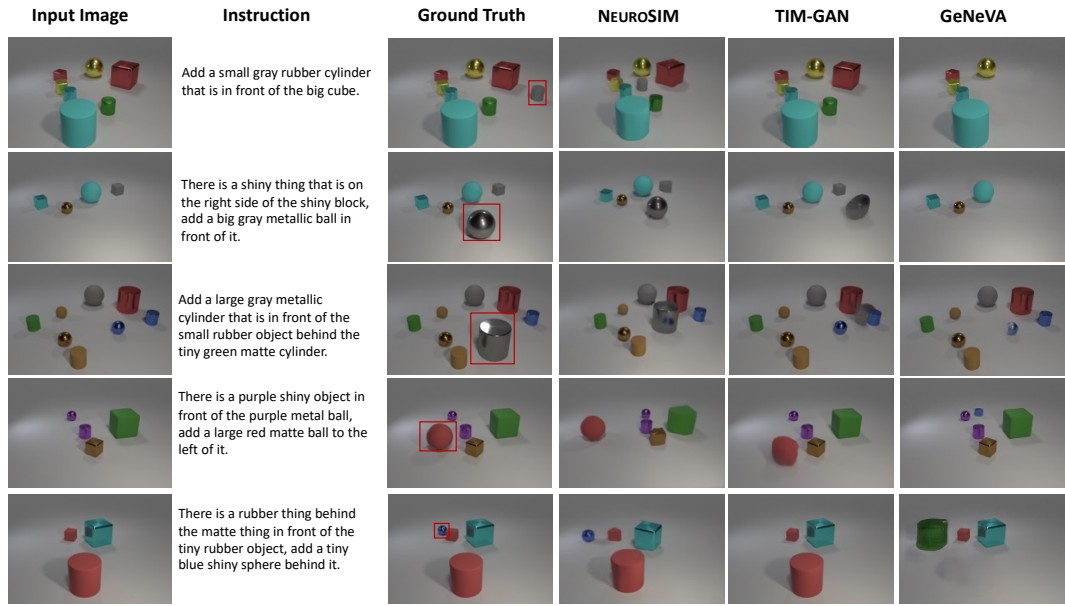

Figure 9: Visual comparison of NEUROSIM with TIM-GAN and GeNeVA for the *add* operator. The red bounding boxes in the ground truth output image indicate the objects required to add to the input image.

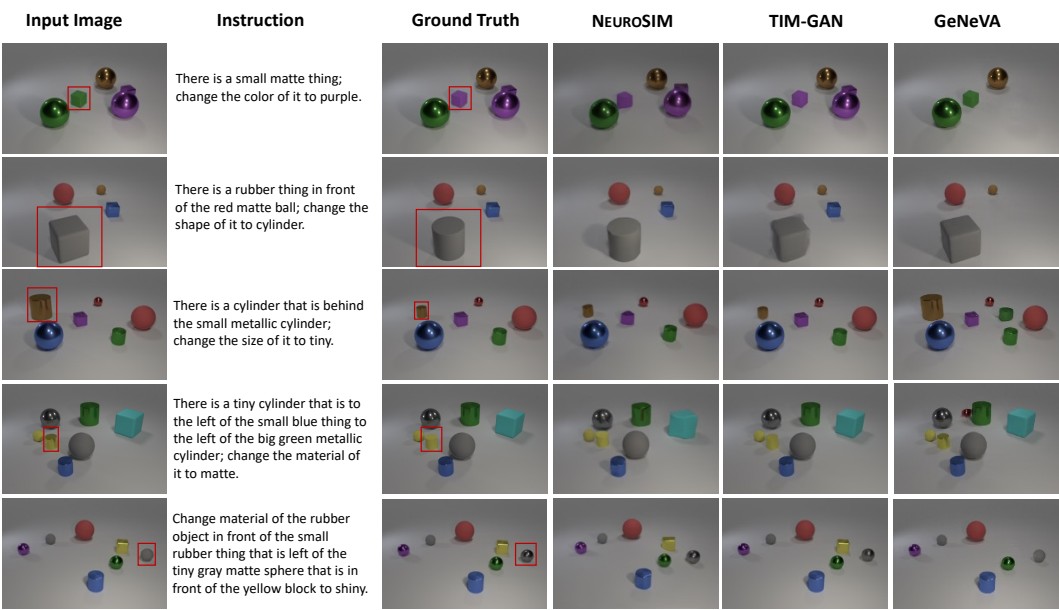

Figure 10: Visual comparison of NEUROSIM with TIM-GAN and GeNeVA for the *change* operator. The red bounding boxes in the input and ground truth output image indicate the objects required to be changed.

by TIM-GAN are better in quality as compared to NEUROSIM. We believe the reason for this is that TIM-GAN, being fully supervised, only changes a small portion of the image and has learnt to copy a significant portion of the input image directly to the output. How ever this doesn't insure the semantic correctness of TIM-GAN's manipulation, as described above with examples where it makes errors. The images generated by NEUROSIM look slightly worse since the entire image is generated from object based embeddings in the scene graph. Improving neural image rendering from scene graphs can be a promising step to improve NEUROSIM.

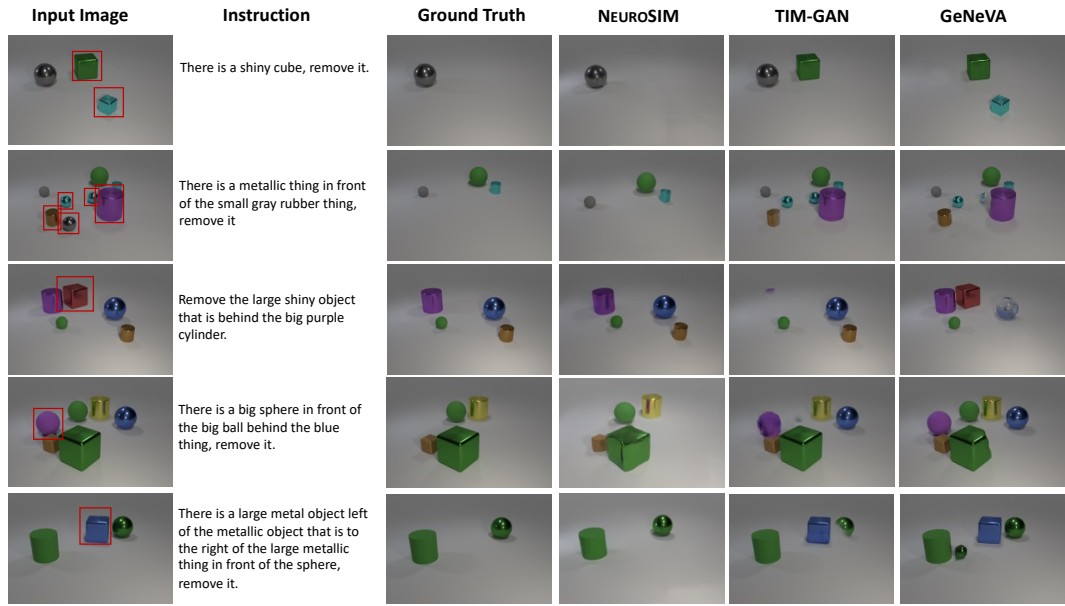

Figure 11: Visual comparison of NEUROSIM with TIM-GAN and GeNeVA for the *remove* operator. The red bounding boxes in the input image indicate objects required to be removed.

## I ERRORS

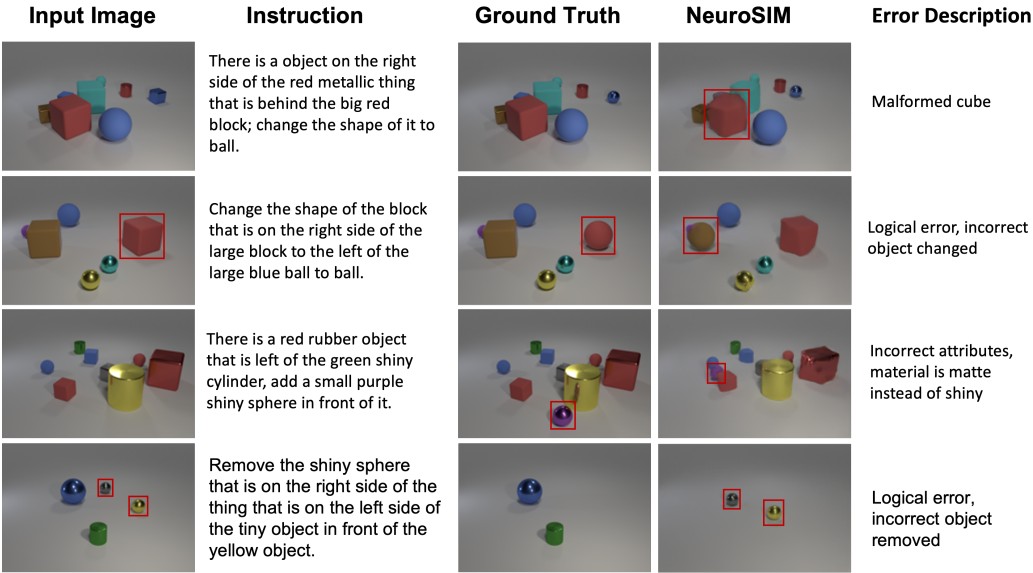

Figure 12: Types of errors in NEUROSIM.

Figure 12 captures the images generated by our model where it has made error. The kind of errors that NEUROSIM makes can be broadly classified into three categories.

- **[Rendering Errors]** This set includes images generated by our model which are semantically correct but suffer from rendering errors. The common rendering errors include malformed cubes, partial cubes, change in position of objects, and different lighting.

- **[Logical Errors]** This set includes images generated by our model which have logical errors. That is, manipulation instruction has been interpreted incorrectly and a different manipulation has been performed. This happens mainly due to an incorrect parse of the input instruction into the program,

or manipulation network not trained to the perfection. For example, *change* network changing attributes which were supposed to remain unchanged.

- **[VQA Errors]** The query networks are not ideal and have errors after they are trained on the VQA task. This in turn causes errors in supervision (obtained from query networks) while training the manipulation networks and leads to a less than optimally trained manipulation network. Also, during inference, object embeddings may not be perfect due to the imperfections in the visual representation network and that leads to incorrect rendering.

## J  INTERPRETABILITY OF NEUROSIM

NEUROSIM allows for interpretable image manipulation through programs which are generated as an intermediate representation of the input instruction. This is one of the major strengths of NEUROSIM, since it allows humans to detect where NEUROSIM failed. This is not possible with purely neural models, that behave as a black-box. Knowing about the failure cases of NEUROSIM also means that it can be selectively trained to improve certain parts of the network (for eg individually training on change instructions to improve the change command, if the model is performing poorly on change instructions). We now assess the correctness of intermediate programs using randomly selected qualitative examples present in Figure 13. Since no wrong program was obtained in the randomly selected set, we find 2 more data points manually, to show some wrong examples.

## K  ABLATIONS

Table 19 shows the performance of NEUROSIM while certain loss terms are removed while learning of the networks. This depicts the importance of loss terms that we have considered. In particular we test the performance of the network by removing edge adversarial loss used by add network (row 2), object adversarial losses for both add and change networks (row 3, 5), self supervision losses used by add network (row 4), cyclic (row 6) and consistency (row 7) losses used by change network.

| Loss | R1 | R3 |
|---|---|---|
| $\ell$ | 45.3 | 65.5 |
| $\ell - \ell_{\text{edgeGAN}}^{add}$ | 43.7 | 66.0 |
| $\ell - \ell_{\text{objGAN}}^{add}$ | 44.3 | 60.2 |
| $\ell - \ell_{\text{objSup}}^{add} - \ell_{\text{edgeSup}}^{add}$ | 44.1 | 57.9 |
| $\ell - \ell_{\text{objGAN}}^{change}$ | 44.9 | 61.5 |
| $\ell - \ell_{\text{cycle}}^{change}$ | 36.5 | 51.1 |
| $\ell - \ell_{\text{consistency}}^{change}$ | 31.0 | 44.8 |

Table 19: Ablations conducted by removing some loss terms. $\ell$ is the total loss before any ablation. For each loss term being removed, the superscript denotes which network it belongs to (add or change). Ablations are conducted for the setting where $\beta = 0.054$ (see main paper Section 4 for the definition of $\beta$)

## L  HUMAN EVALUATION

Table 20 for the questions asked to human evaluators for the human evaluation study. See Section 4.5 of the main paper for more details.

Figure 13: Qualitative examples of generated programs by NEUROSIM.

| | |
|---|---|
| Question 1: | **[Change]** Are all the attributes (color, shape, size, material, and relative position) of the changed object mentioned in the instructions identical between the ground truth image and the system-generated image?

**[Add]** Are all the attributes (color, shape, size, material, and relative position) of the added object mentioned in the instructions identical between the ground truth image and the system-generated image?

**[Remove]** Are same objects removed in ground truth image and the system-generated image? |
| Question 2: | **[Change]** Are all the attributes (color, shape, size, material, and relative position) of the remaining objects identical between the ground truth image and the system-generated image?

**[Add]** Are all the attributes (color, shape, size, material, and relative position) of the remaining objects identical between the ground truth image and the system-generated image?

**[Remove]** Are all the attributes (color, shape, size, material, and relative position) of the remaining objects identical between the ground truth image and the system-generated image? |

Table 20: Questions asked to human evaluators for evaluating NEUROSIM and TIM-GAN. Note that there are some variations in the questions for Change, Add, and Remove instructions dues to different semantic nature of the instructions.

