# OpenReview forum: "Weakly Supervised Neuro-Symbolic Image Manipulation via Multi-Hop Complex Instructions"
_ICLR.cc/2023/Conference — Submitted to ICLR 2023_

### Official Review · Reviewer_zm9X · 2022-10-24

**Confidence:** 5
**Correctness:** 3
**Technical Novelty And Significance:** 3
**Empirical Novelty And Significance:** 3
**Recommendation:** 6

**Clarity, Quality, Novelty And Reproducibility:**

The paper clarified its contributions and approach most clearly.
In general, the paper is novel, can be reproduced and is of fair quality.

**Strength And Weaknesses:**

*[Strength]*
1. This work designed novel neural modules and a training strategy that only uses VQA annotations as weakly supervised data for the task of image manipulation.
2. This work designed evaluation metrics to better evaluate the claimed contribution.

*[Weakness]*
1. What is the performance if there is no VQA training on the visual representation network, semantic parsing module, and concept quantization network (direct end-to-end training on the manipulation task)?
2. What is the performance of the add, remove and change manipulations, respectively?


**Summary Of The Paper:**

This work extends neuro-symbolic approaches for the image manipulation task and proposes a solution referred to as NEUROSIM. Previous work requires supervised training data in the form of manipulated images or can only deal with simple reasoning instructions over single object scenes.
The proposed work performs complex multi-hop reasoning over multi-object scenes and only requires weak supervision in the form of annotated data for VQA by parsing an instruction into a symbolic program, based on a Domain Specific Language comprising of object attributes and manipulation operations.
The proposed method is evaluated on the dataset collected by the author(s) named CIM-NLI.

**Summary Of The Review:**

This paper proposed a method to solve the image manipulation problem and collected new datasets for better evaluate the contribution. Even though there is a lack of some experiments, overall, I'm leaning to accept this paper.

---

> ### Author Response · Authors · 2022-11-14
> **Author Response to Reviewer zm9X [Part 3]**
>
> **Overall performance**
>
> Given the significant increase in performance of NeuroSIM when using supervised data, we also test it’s generalization capability (Analogous to Section 4.2, 4.3), and quality of image retrieval (Analogous to Section 4.5).
>
> **Zero-shot Generalization to larger scenes (Extension of Table 5 of main paper)**
>
> We see that NeuroSIM(e2e) shows improved zero-shot generalization to larger scenes. Even when trained on just 5.4k CIM-NLI data, NeuroSIM(e2e) improves over TIM-GAN-54k by 3.9 R@1 points. A 5.3 point improvement over TIM-GAN is observed when full CIM-NLI data is used.
>
> | Model              | R1   | R3   |
> |--------------------|------|------|
> | TIM-GAN 5.4k       | 30.2 | 80.7 |
> | TIM-GAN 54k        | 66.3 | 92.4 |
> | NeuroSIM 5.4k      | 63.7 | 89.1 |
> | NeuroSIM(e2e) 5.4k | 70.2 | **92.6** |
> | NeuroSIM(e2e) 54k  | **71.6** | 91.7 |
>
>
>
> **Performance with increasing reasoning hops (Extension of Table 3 of main paper)**
>
> (Method with lowest drop (ZH-MH) is marked **bold**)
>
> We see that NeuroSIM(e2e) achieves the lowest drop when compared to TIM-GAN. NeuroSIM(e2e) improves over weakly supervised NeuroSIM baseline by 6.6 R@1 points
>
>
> | Model              | Hops |      |       |
> |--------------------|------|------|-------:|
> |                    | ZH   | MH   | (drop) ZH-MH |
> | TIM-GAN 5.4k       | 56.4 | 41.6 | -14.8 |
> | TIM-GAN 54k        | 84   | 76.2 | -7.8  |
> | NeuroSIM 5.4k      | 64.5 | 63   | **-1.5**  |
> | NeuroSIM(e2e) 5.4k | 69.4 | 67.3 | -2.1  |
> | NeuroSIM(e2e) 54k  | 71.1 | 69.6 | **-1.5**  |
>
>
>
>
> **Quality of scene graph via retrieval (Extension of Table 6 of main paper)**
>
> We see that supervised training significantly improves the scene graph quality, thus improving retrieval performance. Supervised training improves retrieval by 7.3 R@1 points over weakly supervised NeuroSIM baseline
>
>
> | Model         | R1   | R3   |
> |---------------|------|------|
> | Text-Only     | 0.2  | 0.4  |
> | Image-Only    | 34.1 | 83.6 |
> | Concat        | 39.5 | 86.9 |
> | TIRG          | 34.8 | 84.6 |
> | NeuroSIM      | 85.8 | 92.9 |
> | NeuroSIM(e2e) | **93.1** | **96.7** |
>
>
> These new results significantly improve NeuroSIM in comparison to supervised approaches across almost all settings. For further improving the above results, one can finetune the image decoder, and the visual representation network as well, which should significantly improve the results further. This may take a large amount of time for tuning, hence we leave this for the final version of our paper.
>
> We hope that our answers to the reviewers questions, and new results resolves the reviewer’s concern regarding missing experiments. We would like to thank the reviewer again for their questions that we believe have led to a significant improvement in NeuroSIM’s performance and comparisons to supervised baselines. We are working on updating our paper with these new results and discussions and will update the paper soon.
>
> Please let us know if you have any other questions or suggestions.

---

> ### Author Response · Authors · 2022-11-14
> **Author Response to Reviewer zm9X [Part 2]**
>
> **[End-to-End training]**
>
> We believe that by end-to-end training, the reviewer means that we use the ground truth output images as the supervision for our modules. We would like to note that the main objective of the paper has been to make use of weakly supervised VQA data for the manipulation task. An end-to-end training of our network using only manipulation data is a great idea to closely compare the effectiveness of neuro-symbolic models with purely neural models in a purely supervised setting. At the same time, direct end-to-end training requires careful design of a training strategy (for eg. curriculum learning as used by NSCL and backpropagating pixel based loss to the semantic parser) and multiple iterations to perfect.
>
> In order to address the reviewer’s question in the limited time we have, we instead perform a slightly modified experiment which shows the benefit of end-2end training for our network and that it has the capacity to exploit the supervised data effectively. Specifically, we start with our currently trained model using VQA annotations. We then fine-tune it using supervised manipulation data. We refer to this variation as NeuroSIM(e2e) below. Our experiments show that even adding a small amount of supervised data can boost up the performance of our weakly supervised model by a large margin (for eg. ~9-10 points for the change instruction). The detailed results are as follows.
>
>
>
> **Individual (change/add/remove) results (R@1) on varying training data levels (#CIM-NLI are the number of CIM-NLI examples used while training).**
>
>
> **Change Instruction**
>
> | Model         | #CIM-NLI=5.4k | #CIM-NLI=7k | #CIM-NLI=10k | #CIM-NLI=20k | #CIM-NLI=54k |
> |---------------|---------------|-------------|--------------|--------------|--------------|
> | GeNeVA        | 5.9           | -           | -            | -            | 4.1          |
> | TIM-GAN       | 41            | 42.9        | 49.8         | 62.5         | **78.3**         |
> | NeuroSIM      | 57.2          | 57.3        | 57.2         | 57.2         | 57.1         |
> | NeuroSIM(e2e) | **66.2**          | **66.3**       | **66.6**         | **67.4**        | 69.6         |
>
>
>
> **Add Instruction**
>
> | Model         | #CIM-NLI=5.4k | #CIM-NLI=7k | #CIM-NLI=10k | #CIM-NLI=20k | #CIM-NLI=54k |
> |---------------|---------------|-------------|--------------|--------------|--------------|
> | GeNeVA        | 0             | -           | -            | -            | 0.7          |
> | TIM-GAN       | 1.9           | 4.9         | 8.6          | 10.3         | **13.1**         |
> | NeuroSIM      | 4.9           | 6.4         | 5.7          | 5.9          | 5.6          |
> | NeuroSIM(e2e) | **8.8**           | **8.9**        | **9.2**          | **10.5**         | 10.6         |
>
>
> **Remove Instruction**
>
> The results for the remove instruction remain the same since it’s a symbolic operation.
>
>
> | Model         | #CIM-NLI=5.4k | #CIM-NLI=7k | #CIM-NLI=10k | #CIM-NLI=20k | #CIM-NLI=54k |
> |---------------|---------------|-------------|--------------|--------------|--------------|
> | GeNeVA        | 13.2          | -           | -            | -            | 8.7          |
> | TIM-GAN       | 49.6          | 47          | 53.9         | 65.3         | **78.0**           |
> | NeuroSIM      | **69.6**       | **69.5**         | **69.5**       | **69.5**         | 69.6         |
> | NeuroSIM(e2e) | **69.6**        | **69.5**         | **69.5**        | **69.5**         | 69.6         |

---

> ### Author Response · Authors · 2022-11-14
> **Author Response to Reviewer zm9X [Part 1]**
>
> We thank the reviewer for their insightful feedback, and for pointing to our method’s novelties including “novel neural modules”, “training strategy”, and creation of new datasets/evaluation metrics.
>
> Our response to the reviewer’s question:
>
> **[Individual performance of the add, remove and change manipulations, respectively]**
>
> Following are the performance numbers (R@1) for add, remove and change manipulations respectively. Note that you can find a more detailed version (including R@3 scores) in our paper (Appendix Section D.3 and Appendix Table 13) along with a detailed discussion.
>
> |Model    |Instruction Type | β=0.054 | β=0.07 | β=0.1 | β=0.2 | β=0.54 |
> |-----------|--------|---------|--------|-------|-------|--------|
> | Geneva    | add    | 0       | --     | --    | --    | 0.7    |
> |           | change | 5.9     | --     | --    | --    | 4.1    |
> |           | remove | 13.2    | --     | --    | --    | 8.7    |
> | TIM-GAN   | add    | 1.9     | 4.9    | 8.6   | 10.3  | 13.1   |
> |           | change | 41      | 42.9   | 49.8  | 62.5  | 78.3   |
> |           | remove | 49.6    | 47     | 53.9  | 65.3  | 78.0    |
> | Neuro-SIM | add    | 4.9     | 6.4    | 5.7   | 5.9   | 5.6    |
> |           | change | 57.2    | 57.3   | 57.2  | 57.2  | 57.1   |
> |           | remove | 69.6    | 69.5   | 69.5  | 69.5  | 69.6   |

---

> ### Author Response · Authors · 2022-11-18
> **Paper update**
>
> Dear Reviewer zm9X
>
> We have updated our paper based on comments from all reviewers. Among other changes, we have also added the new results obtained for end-to-end training of NeuroSIM.
> Please see our common response titled “Summary of Updates”.

---

> ### Author Response · Authors · 2022-11-25
> **Looking forward to further discussion on our paper**
>
> Dear Reviewer zm9X
>
> Since we have reached the middle of the AC-Reviewer discussion phase and are towards the end of the overall discussion phase, we thought of sending this note since we have not heard back from you yet regarding our response to your comments/questions. We wanted to check if we were able to resolve all your concerns/questions, and if you had any further comments on our work. We will be happy to address any additional concerns that you might have, and we look forward to engaging in further discussions on our paper.

---

> ### Comment · Reviewer_zm9X · 2022-12-13
> **Updated Review**
>
> The authors' review addressed my first concerns. However, after viewing other reviews, I still have concerns:
>
> a. Experiments: The experiments are only limited to one dataset, and CLEVR is a relevantly simple one, which cannot fully validate its contribution.
>
> b. The experiments of GQA and Minecraft are without quantitative results which cannot well-evaluate the model performance.
>
> c. Methods: Previously, I supported this work since I think it proposed a novel methodology that contributes to the community. However, after viewing other reviews and discussions, there are related works in text2image generation that have never been mentioned and compared, reducing its contribution.
>
> In summary, I'm leaning to change the review to **5: marginally below the acceptance threshold**.

---

> > ### Author Response · Authors · 2022-12-13
> > **Author response to updated review**
> >
> > We thank the reviewer for their comment, and we are happy to know that our earlier response could address their original concerns. Regarding the additional concerns raised by the reviewer, we would like to highlight the following.
> >
> > a. CLEVR is a versatile benchmark which has been used for experiments, in several existing works such as [1, 2, 3], published at top venues, which experiment primarily with CLEVR and its extensions, or make use of CLEVR and some other simple artificial datasets. CLEVR has been used for experimentation in a large number of papers, some of which are as listed here (https://paperswithcode.com/dataset/clevr):
> > So, we believe that demonstrating our results on CLEVR based data can be treated as reasonable evidence that our approach has merit compared to existing baselines. We also create another dataset for comparing generalization abilities to larger scenes. At the same time, based on the Reviewer oHcq comments, we did additional experiments on two additional datasets : Minecraft and GQA, as a proof of concept. Next, we address the reviewer’s concerns regarding our results on these datasets.
> >
> >
> > b. We evaluated our method on GQA and Minecraft datasets due to their common use in the literature of visual question answering. For these datasets, we don't have output images as well as a curated dataset of manipulation instructions, hence obtaining quantitative results (recall metrics) was a challenge. However, if quantitative evaluation is needed, we will be happy to curate a set of manipulation instructions and carry out human evaluation studies which will help quantitatively evaluate our method. We will be happy to conduct such a human study and report it in the final version of our paper. Comparison with baselines was not possible also because they require output images as supervision, which is not available to us, and curating such output images may require significantly more (annotation) effort, especially on real data.
> >
> > c. We agree that there has been a surge in text to image generation methods such as using diffusion models (Saharia et. al 2022, Ramesh et. al 2022). We note that **we cite these relevant works in Section 2: Related works (last paragraph) in the main paper**, and also talk about the differences they have with our setting. In particular, image editing through diffusion models requires both the caption of the input image as well as the caption of the output image (Ramesh et. al). It is unclear what this caption will be in the first place, for multi-object scenes working with complex manipulations (caption would require details of multiple objects, and details about their relations with other objects. For e.g., for a 10 object scene, we can imagine that this caption may become very large).  There are other text2image works as well such as Dong et al. (2017), TAGAN (Nam et al., 2018), and ManiGAN (Li et al., 2020), as mentioned by reviewer oHcq in their latest comment. We have cited these works as well in Section 2: Related works (second paragraph), and also mention how the setting is different from ours. In particular, the complexity of their natural language instructions is restricted to 0-hop. Most of their experimentation is limited to single (salient) object scenes, and it is unclear how these strategies would perform with multi-object situations with intricate relationships. In light of this, we have designed a novel neuro-symbolic model for image manipulation via complex multi hop instructions.
> >
> > To the best of our knowledge, we have cited and compared to all the relevant works in our paper. We would request the reviewer to let us know if there are any other text2image works that we haven’t cited or compared to, and we would be happy to do so.
> >
> >
> > **References**
> >
> > 1. Locatello et. al, Object-Centric Learning with Slot Attention. NeurIPS 2020
> >
> > 2. Stammer et. al, Right for the Right Concept: Revising Neuro-Symbolic Concepts by Interacting with their Explanations. CVPR 2021
> >
> > 3. Desai et. al, Probabilistic Neural-symbolic Models for Interpretable Visual Question Answering. ICML 2019

---

> > > ### Author Response · Authors · 2022-12-20
> > > **Following up**
> > >
> > > Dear Reviewer zm9X
> > >
> > > We hope that based on our response, we were able to resolve your concerns regarding the novelty and experimental validation of our work. We would be more than happy to address any follow-up concerns/questions that you may have until the end of the discussion phase.

---

### Official Review · Reviewer_oHcq · 2022-10-26

**Confidence:** 5
**Correctness:** 3
**Technical Novelty And Significance:** 3
**Empirical Novelty And Significance:** 3
**Recommendation:** 6

**Clarity, Quality, Novelty And Reproducibility:**

Clarity and Quality
- The manuscript should be proofread a bit more.
  - Abbreviations such as NSCL and VQA appear suddenly in the abstract without any explanation. Their full names should be stated first.
  - Only the parentheses for CIM-NLI are in italics.
  - All paragraphs are long; it is difficult to grasp the structure of the text. Paragraph writing should be kept in mind.

Novelty
- The third weakness above is one about novelty. Technical novelty is not necessarily high.

Reproducibility
- There is also a detailed description in the appendix, and the code is provided. Reproducibility is sufficient.


**Strength And Weaknesses:**

Strengths
1. Efficient method utilizing neuro-symbolic approach for image manipulation through text.
1. A novel dataset and the source code is provided.

Weaknesses
1. This is a common issue for studies that use synthetic data, such as CLEVR, but operating in such a controlled environment does not guarantee that it will actually work on real images as expected in an application. Although this paper states that multi-hop reasoning is required as "complex image manipulation," the proposed method would also need to be evaluated on real images, as done for NS-CL (Mao et al., 2019).
1. While the proposed method does not require edited images, it requires annotation for VQA and Domain Specific Language (DSL) design. Moreover, its effectiveness has only been confirmed on controlled CLEVR-based datasets.
1. This paper is close to system integration. It is certainly the first neuro-symbolic approach to text-guided image editing, but it is a combination of known ideas when broken down into its components. For example, the introduction of the symbolic approach to Vision & Language, the use of VQA for image generation, and the generation of scene graphs from images and images from scene graphs in the method are all existing ideas.

**Summary Of The Paper:**

This paper addresses image manipulation through text. The proposed method is based on an existing method (Mao et al., 2019) proposed for Visual Question Answering (VQA) with a neuro-symbolic approach. The proposed method leverages VQA annotations and shows better accuracy than comparative methods with weakly supervised learning. Experiments are conducted on a synthesized dataset utilizing CLEVR.

One note: Although one could imagine weakly supervision by VQA could utilize the errors from VQA in the image generation network (Niu et al., 2020), the proposed method simply trains some modules using VQA annotations.

Tianrui Niu, Fangxiang Feng, Lingxuan Li, and Xiaojie Wang. Image Synthesis from Locally Related Texts. International Conference on Multimedia Retrieval, 2020.

**Summary Of The Review:**

Overall, the reviewer is on the borderline but slightly leans toward rejecting this paper. A response from the authors regarding the above weaknesses could improve the score.

----

**Updated review**

The authors have answered almost all of the reviewers' concerns. In particular, the addition of experiments beyond the CLEVR dataset, showing the strength of the proposed method for synthetic data and the challenges and prospects for real data, will contribute to subsequent research.

Therefore, the reviewer improves the review score by one. However, the cause of the reviewer's still borderline judgment is a concern related to the following comments from the authors.

> Moreover, it is not clear as to how we can solve the task of complex image manipulation without using either explicit supervision in the form of manipulated images or VQA examples.

As cited by the authors, Dong et al. (2017), TAGAN (Nam et al., 2018), and ManiGAN (Li et al., 2020) also propose image manipulation with weak supervision. These methods learn image manipulations from image-caption pairs. Regarding ease of data collection, these image-caption pairs are more effortless than VQA.

The authors claim that these studies do not cover multi-hop operations on images containing multiple objects. However, some papers have experimented with a dataset, COCO, containing multiple objects, and whether the methods proposed by these papers are 0-hop remains a matter of hypothesis. These methods have the possibility to generate proper images even if an instruction includes multi-hop instructions.

Moreover, the authors' last point is whether these conventional methods can output images with only difference (delta) instructions for image manipulation. For example, however, Fig. 6 of ManiGAN (Li et al., 2020) shows that it is possible to manipulate images even from a simple instruction that looks like a delta.

The image manipulation performed by the authors on real image data is also low as its subjective realization. It would have been necessary to directly compare these methods to the proposed one on a common dataset.

---

> ### Author Response · Authors · 2022-11-14
> **Author Response to Reviewer oHcq [Part 2]**
>
> **[Regarding Novelty]**
>
> While on the first look, our proposed NeuroSIM system may give an appearance as being an integrated system, we would like to emphasize that there are several non-trivial and novel aspects to it. To bring across our point, please allow us to reiterate them here.
>
> **Novelty of the Problem Set-up:** We believe that posing the task of image manipulation requiring zero annotations of manipulated images in itself is novel as well as very useful. To the best of our knowledge, we have not come across any work in this direction. Moreover, the fact that our problem setting has multi-object scenes and complex instructions of different types further enhances our contribution. The highlight is using weak supervision from VQA to solve this task because that brings down
> the annotation cost by a significant margin.
>
>
> **Novelty in Methodology:**  We design an effective neuro-symbolic approach for the task. Although a neuro-symbolic approach to “reasoning” tasks like VQA has already been explored, it hasn’t been explored in a setting like ours that requires “manipulation” as well as “reasoning”. To address this challenge, we develop novel neuro-symbolic manipulation modules (add, remove, change), and propose novel loss functions to train them from VQA based weak supervision. Combining together the existing reasoning modules, newly designed manipulation modules, along with well known concepts of scene graph to image generation, is a novelty of our methodology.
>
> **Novelty in Experimentation:**
> We  provide an extensive set of experiments (Section 4.1 to 4.4) demonstrating the capability of our model to surpass SOTA supervised baselines, in low data setting, for multi-hop image manipulation. We also demonstrate effective zero shot generalization to out-of-distribution samples (CIM-NLI-Large), and show that NeuroSIM performs almost equally on Zero Hop vs Multi Hop instructions. Prior works on text guided image manipulation [2, 3] only measure basic in-distribution performance.
> Our study provides a systematic way of benchmarking performance of future models given that we have created new datasets for it, and puts light on various limitations of neural models. We feel this significantly adds to our novelty.
>
> So, overall, we do believe that our work is novel in various aspects, and opens up possibilities of further research on using VQA style reasoning datasets for achieving more complex manipulation tasks over images.
>
> **[Comments on clarity and quality]**
>
> Thank you for the suggestions to improve clarity. We are working on this aspect by improving the structure of the text and will update the changes along with the first revised version of the paper that we are planning to upload soon. If the reviewer has in mind some specific portions of the paper that can be especially improved or are unclear, we would be happy to focus on them first.
>
>
> **References**
>
> [1] Ellis et. al, DreamCoder: bootstrapping inductive program synthesis with wake-sleep library learning, PLDI 2021
>
> [2] El-Nouby et. al, Tell, Draw, and Repeat: Generating and modifying images based on continual linguistic instruction, ICCV 2019
>
> [3] Zhang et. al, Text as neural operator: Image manipulation by text instruction. ACM MM
>
> [4] Ellis, et. al, Library learning for neurally-guided bayesian program induction. NeurIPS 2018
>
> [5] Yi, et. al,Neural-Symbolic VQA: Disentangling Reasoning from Vision and Language Understanding. NeurIPS 2018
>
> [6] Mao, et. al, The Neuro-Symbolic Concept Learner: Interpreting Scenes, Words, and Sentences From Natural Supervision. ICLR 2019
>
> [7] Yi, et. al, CLEVRER: CoLlision Events for Video REpresentation and Reasoning. ICLR 2020
>
> [8] Jiang, et. al, Language as an Abstraction for Hierarchical Deep Reinforcement Learning. NeurIPS 2019

---

> ### Author Response · Authors · 2022-11-14
> **Author Response to Reviewer oHcq [Part 1]**
>
> We thank the reviewer for their insightful comments and for describing our contribution as an effective neuro symbolic approach for the image manipulation task.
>
> Our response to the reviewer’s questions and comments:
>
> >One note: Although one could imagine weakly supervision by VQA could utilize the errors from VQA in the image generation network (Niu et al., 2020), the proposed method simply trains some modules using VQA annotations.
>
> We note that using VQA supervision will require having an annotated dataset which has both VQA question-answers, as well as associated manipulation commands. In our case, we use a separate VQA dataset, and during training of manipulation networks, we don’t use any question as input, rather we get supervision from modules trained using the VQA dataset. We thank the reviewer for the pointer to the relevant paper by Niu et al. [2020]. We will be adding a citation and discussion in our revised version.
>
> **[Requirement of annotation for VQA and DSL]**
>
> We consider it as our strength that we can perform the task of complex image manipulation  “just using VQA annotations”. The cost of VQA annotations would be much smaller than getting full manipulated image level supervision (see Section 4 in the paper). Moreover, it is not clear as to how we can solve the task of complex image manipulation without using either explicit supervision in the form of manipulated images or VQA examples.
>
> Although we require a DSL, It can be seen both as a strength and a limitation. It is a strength that we can incorporate domain specific information through the DSL that purely neural models don’t allow for, and this leads to NeuroSIM requiring no supervised image level data for training the models. On the flip side, one can also view DSL as a limitation. However, some  recent works [1,4] have explored automatic learning of the DSL for different domains, and extending this to our setting is an interesting future direction for our work.
>
> **[Experimenting with CLEVR based datasets]**
>
> While we agree that a natural and more exciting extension of our work would be to evaluate on real image datasets, it should not undermine our primary contribution of proposing the first-of its kind set up for the task of weakly supervised image manipulation via complex instructions. We have used CLEVR based dataset(s) to demonstrate the effectiveness of our proposed approach under weak supervision vis-a-vis purely supervised models, in a controlled setting. Such a setting also highlights the limitations of neural models, especially when generalizing to out-of-distribution samples (e.g., training on CIM-NLI, and testing on CIM-NLI-large). Earlier, CLEVR and its extensions have been used to test the effectiveness of multiple other works for related problems such as the first neuro-symbolic VQA model (NS-VQA [5]) , NSCL[6],  and other works [7, 8], etc.
>
> One of the main challenges that we face while experimenting on real data is: to the best of our knowledge, there is **no such existing real dataset** with complex manipulation commands, which provides annotations in the form of (image, manipulation-command) pairs, leave aside the additional annotation in the form of target image (which is required by purely neural models).
>
> Further, though we agree with the reviewer that NSCL [6] have evaluated their approach on real image datasets, a close reading of their work shows that although very useful, it has been done in a carefully controlled setting. For instance, they assume availability of a pre-trained syntactic dependency parser, when working with real data. This points to the possible need for making additional assumptions (in terms of availability of pre-trained models) when working with real data, and hence, in our case as well, we might be able to show the experiments on real data in a controlled setting.
>
> Specifically, we are currently working on extending a real world dataset to incorporate a subset of the manipulation operations, so that we can at least show a proof-of-concept demonstration of the effectiveness of our approach in a controlled setting. We hope to share some results soon in this regard and we appreciate the reviewer's patience in this matter. Finally, we would like to note that due to the non-availability of output manipulated images, a comparison with purely neural approaches would not be possible in any case.

---

> ### Author Response · Authors · 2022-11-18
> **Dataset related comments + Paper update**
>
> Dear Reviewer oHcq
>
> Thank you for your patience
>
> Based on your comments, we experimented with two datasets, an extension of Minecraft dataset used in NS-VQA[1] and GQA[2] dataset. Given the limited time, we are able to show preliminary results which show that our approach can generalise to these datasets. We hope these results can resolve your comments regarding the approach being extended beyond CLEVR based datasets. If you have any comments or questions regarding these experiments, we would be happy to discuss it now or during the next phase of discussion
> We have also resolved your comments on the clarity and readability of the paper.
>
> We have updated the paper based on comments from all reviewers. Please see our common response titled “Summary of Updates”.
>
> **References**
> 1. Yi. et. al, Neural-symbolic VQA: Disentangling reasoning from vision and language understanding. NeurIPS, 2018.
> 2. Hudson. et. al, GQA: A new dataset for real-world visual reasoning and compositional question answering. CVPR, 2019.

---

> ### Author Response · Authors · 2022-11-25
> **Looking forward to further discussion on our paper**
>
> Dear Reviewer oHcq
>
> Since we have reached the middle of the AC-Reviewer discussion phase and are towards the end of the overall discussion phase, we thought of sending this note since we have not heard back from you yet regarding our response to your comments/questions. We wanted to check if we were able to resolve all your concerns/questions, and if you had any further comments on our work. We will be happy to address any additional concerns that you might have, and we look forward to engaging in further discussions on our paper.

---

> ### Author Response · Authors · 2022-12-13
> **Author Response to Updated Review**
>
>
> We thank the reviewer for their updated review and comments, and the fact that they have decided to increase the score based on our response, which addressed their original concerns. Regarding additional concerns raised, we would like to mention the following points:
>
> 1. Our intent to write the following comment
>
> >Moreover, it is not clear as to how we can solve the task of complex image manipulation without using either explicit supervision in the form of manipulated images or VQA examples.
>
> was to stress that some kind of (weak-)supervision would be needed to perform the task of image manipulation, either in the form of output image supervision, VQA based weak supervision, or possibly other forms of weak supervision (as pointed out by the reviewer) of similar complexity. We are sorry if there was any mis-understanding in this regard.
>
> 2. We mentioned 0-hop, in the context of our setting where hops are defined by the number of relational reasoning steps the model has to perform over the multiple objects present in the scene. We argue that to perform such reasoning, captions themselves would have to capture complex dependencies among objects, again requiring significant additional labeling effort. To take an example from Fig. 6 in the ManiGAN paper, as cited by the reviewer, none of the instructions refer to relative positions of various objects in the text (multi-hop reasoning). For example, one of their commands/captions is “evening” (second last column), which changes the entire surrounding to evening.
>
> 3. On the other hand, in our case, we work with significantly more complex relative reasoning such as “Change the color of the yellow object on top of the blue cube which is next to the red sphere to green”. We argue that even to describe such a command requires significant labeling effort in terms of understanding the scene, and relationships between objects, which would be tantamount to doing full VQA. Nevertheless, we agree with the reviewer that it is still a hypothesis, and additional experimentation may be needed to verify this thesis. If the reviewer has a specific experiment in mind for a direct comparison, we will be happy to consider it for adding to our final camera ready version.
>
> 4. Finally, it is unclear if ManiGAN like methods can always work with just “delta” in the image. It is true that Fig 6 in their paper has an example (e.g., one with “Evening” caption) which works with delta, but on the other hand, others (such as “zebra, sand”) require fuller description (eg, no example is shown to edit a particular zebra) and we believe to edit a particular zebra, additional description may be needed capturing relative context, and the caption would become more complex as the number of zebras increases in the image. As argued above, we believe that for more complex commands involving more relative positions of objects, a delta may not always be sufficient, but verifying this is a part of further experimentation.
>
> 5. _“The image manipulation performed by the authors on real image data is also low as its subjective realization. It would have been necessary to directly compare these methods to the proposed one on a common dataset.”_ : -- Our focus in this paper was to work with complex multi-hop instructions, and comparing with existing approaches on real datasets via such instructions, would require generating such a dataset of manipulation instructions and complex captions first (there is none on real data to the best of our knowledge) and also VQA annotations, but we would be happy to look into curating such a dataset in the final version, if so desired. We would also be happy to consider any additional inputs that the reviewer might have in this regard.

---

> > ### Author Response · Authors · 2022-12-20
> > **Following up**
> >
> > Dear Reviewer oHcq
> >
> > We hope that based on our previous response, we were able to resolve your concerns regarding appropriate comparison with prior work, and the novelty of our formulation compared to existing work. We will be happy to address any additional concerns/questions that you might have until the end of the discussion phase.
> >
> > There is one additional note that we forgot to make in the previous response, which may further help address your concern. We would like to _highlight_ that apart from the differences with previous methods mentioned in our previous response, caption to image based approaches such as Dong et al. (2017), TAGAN (Nam et al., 2018), and ManiGAN (Li et al., 2020) have been designed to primarily work with simple change based descriptions, and it is not clear if they can be directly extended to add/remove commands (ref. Table 1 in our main paper, and also abstract of the ManiGAN paper where they only talk about changing attributes). This is also evident from the examples shown in the above papers. We believe this is an important difference where unlike previous work, we can handle change, as well as, add and remove instructions, and show experimental validation for the same.

---

### Official Review · Reviewer_nYyA · 2022-10-28

**Confidence:** 3
**Clarity, Quality, Novelty And Reproducibility:** I have no other concerns.
**Correctness:** 3
**Technical Novelty And Significance:** 4
**Empirical Novelty And Significance:** 4
**Recommendation:** 8

**Strength And Weaknesses:**

This paper is the first work that can handle multiobject scenes with complex instructions requiring multi-hop reasoning, and solve the task without any output image supervision.

**Summary Of The Paper:**

This paper presents an neuro-symbolic, interpretable approach NEUROSIM to solve image manipulation task using weak supervision of VQA annotations, building on existing work on neuro-symbolic.



**Summary Of The Review:**

The overall contribution of this paper is significant.

---

> ### Author Response · Authors · 2022-11-14
> **Author Response to Reviewer nYyA**
>
> We thank the reviewer for their positive comments and describing our contributions as significant.
>
> If the reviewer has any suggestions for further improving our work, please let us know and we would be happy to incorporate them.

---

> ### Author Response · Authors · 2022-11-18
> **Paper update**
>
> Dear Reviewer nYyA
>
> We have updated our paper based on comments from all reviewers.
> Please see our common response titled “Summary of Updates”.

---

> ### Author Response · Authors · 2022-11-25
> **Looking forward to further discussion on our paper**
>
> Dear Reviewer nYyA
>
> We thought of sending a quick note to check if you had any further comments on our paper and/or response that we posted addressing concerns raised by various reviewers. We will be happy to address any additional questions/concerns that you might have.

---

### Author Response · Authors · 2022-11-14
**Common Response from Authors**

We thank all the reviewers for appreciating our work and their insightful comments to further improve our paper. We have tried to address each of the concerns/questions raised, in the response box for respective reviewers. If there are any further questions/comments from any of the reviewers, we will be happy to address them.

We are working on updating our paper with new results and discussions based on reviewers’ concerns, we will update the paper soon.

---

### Author Response · Authors · 2022-11-18
**Summary of Updates**

We thank the reviewers for their insightful questions and comments, for mentioning our novelties and for describing our work as significant and an effective approach.
We provide the following list of changes we have updated our paper with, based on comments and questions from the reviewers.

**[New Results]**

**End to End training**

Addressing reviewer zm9X's comments, Appendix Section E contains experiments to analyze the performance boost that can be obtained by utilizing ground truth output (manipulated) images as the supervision for NeuroSIM's modules after End-to-End training.

This subsection also investigates the effect of supervised data on NeuroSIM's generalization capability to larger scenes, zero vs multi hop performance and retrieval scores for measuring quality of scene graphs.

 These changes can be seen in blue color for easy readability

**Results on datasets other than CLEVR**

Addressing reviewer oHcq's comments, Appendix Section D.5 titled "Results on datasets from different domains" has been added to include NeuroSIM's results on different domains : Minecraft [1] and proof of concept results on Real world GQA [2] dataset. Please see the following sections for details.

a. Subsection D.5.1 contains results on the Minecraft dataset. This dataset contains scenes and objects from the Minecraft game video series and is used in prior works for testing Neuro-Symbolic VQA systems like NSCL and NS-VQA [1]. The setting of the Minecraft worlds dataset is significantly different from the CLEVR in terms of concepts and attributes of objects and visual appearance.

b. Subsection D.5.2 contains results on real-world images from GQA [2] dataset. This dataset was originally used for benchmarking the task of VQA, and contains real world scenes having multiple objects, with different concepts and attributes.

We hope this resolves reviewer oHcq’s concerns regarding experiments with other datasets beyond CLEVR.

These changes can be seen in blue color for easy readability

**[Clarity of Paper]**

As suggested by reviewer oHcq, for improving the clarity and readability of the paper, we have
included the full names of the abbreviation NSCL and VQA in the abstract and resolved the comment regarding CIM-NLI. Other typos are also corrected. These changes can be seen in blue color.

We have performed minor restructuring of the text at a few places like paragraph 1 of Section 1, paragraph 2 of Section 2 and Section 3.3.
If reviewers have any specific comment about the writing that they feel needs to be addressed, we would be happy to incorporate it.


**[General comments]**

We thank the reviewers for their helpful feedback and questions that we feel have helped us to significantly improve our paper, by strengthening NeuroSIM’s results (after end-to-end training) and increasing its scope to other datasets. We hope we were able to answer all questions posed by the reviewers.
Since we are towards the end of the first phase of discussion, we hope to hear back from reviewers, and look forward to answering any other questions now or during the next phase of discussion.

**References**

1. Yi. et. al, Neural-symbolic VQA: Disentangling reasoning from vision and language understanding. NeurIPS, 2018.
2. Hudson. et. al, GQA: A new dataset for real-world visual reasoning and compositional question answering. CVPR, 2019.

---

### Author Response · Authors · 2022-11-25
**Looking forward to further discussion on our paper**

Dear Area Chair,

Since we have reached the middle of the AC-Reviewer discussion phase and are towards the end of the overall discussion phase, we wanted to know if we have resolved the reviewers’ questions and comments, and if they are satisfied with our pointwise response to each of the questions/concerns raised by them. We thought of sending this note, since we have not yet heard back from any of the reviewers regarding our response to their questions/concerns. We look forward to engaging in fruitful discussions on our paper, and resolving any further questions/concerns that reviewers might have.

---

### Decision · Program_Chairs · 2023-01-20

**Decision:**

Reject

**Justification For Why Not Higher Score:**

During the AC-reviewer discussion, two critical concerns were raised:
- The experimental results are not detailed and only reported on one single dataset, CLEVR. Although the Appendix shows some qualitative results on other datasets, no quantitative results are reported, which raises a major concern for the generalization of the proposed method.
- The novelty may be limited. The paper lacks discussion and comparison with recent work on iterative text-to-image editing/generation.

Therefore, the AC and the reviewers reach an consensus that this paper is not ready for publication in its current form. Both Reviewer oHcq and zm9X stated that they are willing to lower their scores to 5. So we would like to recommend a reject.

**Justification For Why Not Lower Score:**

N/A

**Metareview: Summary, Strengths And Weaknesses:**

This paper proposes a Neuro-Symbolic approach for image editing with text instructions, based on previous neuro-symbolic method for VQA. The authors present the results on the CLEVR dataset. The method itself is interesting and Neuro-Symbolic approaches are rarely explored in the image editing domain. However, the reviewers and the AC raised critical concerns for this work, especially during the AC-reviewer discussion: (1) the experimental results are not detailed and only reported on one single dataset, CLEVR. Although the Appendix shows some qualitative results on other datasets (Minecraft and GQA), no quantitative results are reported, which raises a major concern for the generalization of the proposed method. (2) the paper lacks essential discussion and comparison with recent work on iterative text-to-image editing/generation. Therefore, the reviewers and the AC reach a consensus that this paper is not ready for publication in its current form.

**Summary Of Ac-Reviewer Meeting:**

The AC-reviewer meeting was very helpful and included the AC, Reviewer oHcq, and Reviewer zm9X.
Reviewer nYyA gave an extremely short review with a rating of 8, who also did not participate in the meeting or the online discussion. So we believe Reviewer nYyA's review should be discounted.

During the discussion, two critical concerns were raised:
- The experimental results are not detailed and only reported on one single dataset, CLEVR. Although the Appendix shows some qualitative results on other datasets (Minecraft and GQA), no quantitative results are reported, which raises a major concern for the generalization of the proposed method.
- The paper lacks essential discussion and comparison with recent work on iterative text-to-image editing/generation.

Therefore, the AC and the reviewers reach a consensus that this paper is not ready for publication in its current form. Both Reviewer oHcq and zm9X stated that they are willing to lower their scores to 5. So we would like to recommend a reject.